# IMPLICIT REGULARIZATION THROUGH HIDDEN DIVERSITY IN NEURAL NETWORKS

## ABSTRACT

A significant body of work has focused on studying the mechanisms behind the implicit regularization observed in neural networks. Recently, developments in ensemble theory have demonstrated that, for a wide variety of loss functions, the expected risk of the ensemble can be decomposed into a bias and variance term together with an additional term called *diversity*. By using this theoretical framework and by interpreting a *single* neural network as an ensemble, we expose a hidden diversity term in the decomposition of a neural network's expected risk. We argue that the additional diversity term regulates the variance error, thus identifying a new source of *implicit regularization* in neural networks. We demonstrate this regularization on regression and classification datasets by estimating the bias, variance, and diversity terms for MLPs and CNNs. Using double descent as an example, we observe that diversity significantly increases for wide overparameterized neural networks. These results demonstrate a new perspective on implicit regularization in neural networks and open new possible avenues of research into their generalization.

## 1 INTRODUCTION

In the overparameterized regime, neural networks seem to defy conventional wisdom: despite the ability to interpolate their training data, neural networks are able to still generalize well on unseen data. Examples of this phenomenon range from fitting neural networks on noisy data (Neyshabur et al., 2015; Zhang et al., 2017) to the classical double descent experiments (Belkin et al., 2019; Nakkiran et al., 2020).

To explain this phenomenon, a long-standing conjecture has been that neural networks experience a form of implicit regularization (Neyshabur et al., 2015; Zhang et al., 2017; Vardi, 2022). The most prominent approach to understanding this implicit regularization has been through analysis of the optimization process, in particular, how gradient descent finds minima in the loss landscape that leads to good generalization for both linear and nonlinear networks (Neyshabur et al., 2015; Gunasekar et al., 2017; Arora et al., 2019; Razin & Cohen, 2020; Li et al., 2021; Lyu & Li, 2020; Chizat & Bach, 2020; Vardi & Shamir, 2021) or modifications to the gradient descent trajectories in the loss landscape (Barrett & Dherin, 2021; Smith et al., 2021). An alternative approach to understanding over-parameterization behavior from a bias-variance decomposition perspective, is by identifying sources of additional randomness to provide a more fine-grained decomposition of the variance error (Geman et al., 1992; Neal et al., 2019; Adlam & Pennington, 2020; D'Ascoli et al., 2020).

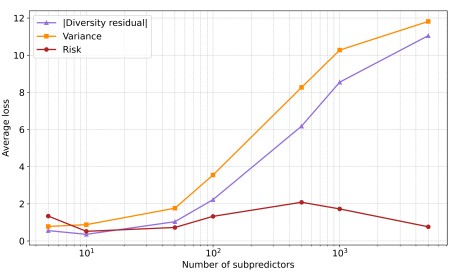

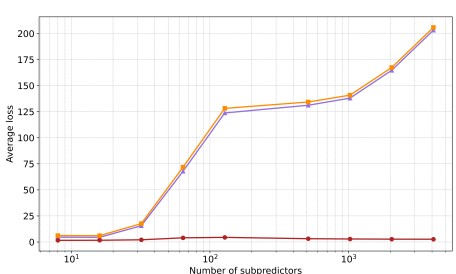

Figure 1: **Implicit regularization** In a new decomposition, variance error increases as parameters increase; a corresponding increase in diversity causes the risk to reduce in the over-parameterized regime. (Top): One-layer MLP trained with full batch SGD on MNIST. (Bottom): Three-layer CNN trained with mini-batch SGD on CIFAR10.

Inspired by the ability of ensembles to reduce overfitting, we view a single neural network as an implicit ensemble. Then, by only considering the randomness introduced by the sampling of training sets, we introduce a new bias-variance decomposition for neural networks that exhibit an additional term that acts as an implicit regularizer. Our main contributions are as follows:

- Development of a theoretical framework to analyze neural network behavior in a regression or classification setting. This framework is applicable to a wide range of loss functions and any model that contains a dense layer.

- Exposing a new source of implicit regularization in such networks: By viewing a network as an ensemble of subpredictors, the diversity of ensemble members is shown to act as an implicit regularizer.

- We confirm our results empirically by deriving and estimating the bias, variance and diversity terms for two loss functions, namely, square loss and cross entropy. We use several well-known classification and regression datasets and present results for scenarios such as double descent and small/large data limits. In addition, we demonstrate the difference in behavior between networks parameterized with standard and mean-field parameterization (MFP).

## 2 RELATED WORK

Related works that view a neural network as a form of implicit ensemble typically aim to achieve different goals than those studied here. For example, Veit et al. (2016) use an ensemble perspective to study the vanishing gradient problem in deep residual networks. Dropout can be interpreted as training many "thinned-out" subnetworks of the neural network (Srivastava et al., 2014). More recently, when studying continual learning and the problem of catastrophic forgetting, a neural network can be viewed as an ensemble of its weights (Benjamin et al., 2024).

In contrast, Olson et al. (2018) is the only work that we are aware of that studies implicit regularization from an ensemble perspective, and is the work that is conceptually closest to ours. By using a linear program procedure and constraints, they decompose a single neural network into an ensemble of non-overlapping subnetworks under the constraint that the networks have low bias and low subpredictor correlation. (While they refer to this concept as "diversity", this is not the same definition of diversity as ours, and seems to be a proxy for this concept.) In analogy to the implicit regularization seen in random forests, they argue that there is a similar mechanism of implicit regularization for overparameterized neural networks, but this mechanism is not explained. In contrast, in our work we use the recent theoretical framework of Wood et al. (2023) to define the subnetworks in the implicit ensemble, producing results that are more generally applicable, in particular, since our subnetworks are allowed to overlap in terms of parameter sharing. We also use the "unified" notion of diversity of Wood et al. (2023), which exposes the exact mechanism of implicit regularization that was speculated to exist in Olson et al. (2018).

More distantly related to our work are the approaches to implicit regularization discussed in the introduction to this paper. In particular, these works approach the problem from the optimization point of view by studying the implicit bias of gradient descent towards favoring solutions that generalize well (Vardi, 2022; Gunasekar et al., 2017; Arora et al., 2019; Razin & Cohen, 2020; Li et al., 2021; Lyu & Li, 2020; Chizat & Bach, 2020; Vardi & Shamir, 2021; Barrett & Dherin, 2021; Smith et al., 2021). These works do not address implicit regularization through internal structure. Finally, our work crucially depends on the framework of Wood et al. (2023), which was developed for explicit ensembles and as a unifying framework for the notion of diversity. In contrast, our theoretical results extend this framework to the internal structure of a single neural network and, additionally, show that the framework provides new insight into the problem of overparameterized neural networks and overfitting.

## 3 PRELIMINARIES AND BACKGROUND

Crucial to our work is the development of the theoretical framework in Wood et al. (2023) for ensembles, which we briefly review here. In contrast to the decomposition in Geman et al. (1992) for a single model, this theoretical framework offers a novel decomposition of the expected risk of

ensembles into three terms, namely, a bias, variance, and diversity term. All proofs for this section can be found in Appendix A.2.

### 3.1 ENSEMBLES

We define an explicit ensemble to consist of *subpredictors* $\{q_{(i)}\}_{i=1}^m$, where each $q_{(i)} : \mathcal{X} \subseteq \mathbb{R}^{d_i} \to \mathcal{Y} \subseteq \mathbb{R}^{d_f}$. We consider supervised learning and, thus, define a training set $\mathcal{D} = \{(x^{(i)}, y^{(i)})\}_{i=1}^n$, where $x^{(i)} \in \mathcal{X}$, $y^{(i)} \in \mathcal{Y}' \subseteq \mathbb{R}^{d_f}$ (where $\mathcal{Y}'$ may or may not be the same as $\mathcal{Y}$) and $(x, y) \overset{\text{iid}}{\sim} P(X, Y)$. Each subpredictor $q_{(i)}$ is trained using a learning algorithm and a loss function $\ell : \mathcal{Y} \times \mathcal{Y} \to \mathbb{R}^+$ by minimizing the *empirical risk* $R_{emp}[q_{(i)}] = \frac{1}{n} \sum_{j=1}^n \ell(q_{(i)}(x^{(j)}), y^{(j)})$. To emphasize the dependence on the training dataset $\mathcal{D}$, we will sometimes write the output of subpredictor $q_{(i)}$ for a given $x$ as $q_{(i)}(x; \mathcal{D})$.

Depending on the task, the ensemble of subpredictors is then aggregated in some fashion to produce a single output in $\mathbb{R}^{d_f}$. For example, in the case of univariate regression, a popular way to combine the outputs of the ensemble of subpredictors is by a simple average: $\bar{q}(x) = \frac{1}{m} \sum_{i=1}^m q_{(i)}(x)$.

### 3.2 CENTROID AND ENSEMBLE COMBINER

An important aspect of the framework of Wood et al. (2023) is the notion of the *centroid of a distribution*, which was first defined in James & Hastie (1997). Using an equivalent definition of the variance of a random variable $T$ given by $\text{var}(T) = \min_z \mathbb{E}_T[\ell(z, t)]$, where $\ell(z, t) = (z - t)^2$, the nonrandom number $\mathring{t}$, which minimizes the variance, is called the centroid of the distribution. The utility of this formulation is that we can generalize it to other loss functions $\ell$.

**Definition 1** (Centroid of a distribution). *Let $T$ be a random variable. Then, for a given loss $\ell$, we define the* centroid of the distribution $\mathring{t}$ *to be the minimizer of the expected loss over $T$*

$$\mathring{t} = \arg \min_z \mathbb{E}_T[\ell(z, t)],$$

*where $z$ is nonrandom.*

Intuitively, as described in (James & Hastie, 1997), the quantity $\text{var}(T)$ can be interpreted as a measure of the expected distance (in terms of the loss) of the random variable $T$ from its nearest nonrandom number $\mathring{t}$. Importantly, the loss function determines the form of the centroid. We provide the following two lemmas for the square loss function and the Kullback-Leibler divergence (KL-divergence) (Heskes, 1998).

**Lemma 1** (Centroid for least squares loss). *Let $\ell$ be the least squares loss function $\ell(z, t) = (z - t)^2$. Then, for a random variable $T$, the centroid of the distribution is given by*

$$\mathring{t} = \mathbb{E}_T[t]. \tag{1}$$

**Lemma 2** (Centroid for KL-divergence). *For a target probability density $z(y)$, let $f(y)$ be an estimator of this density. Suppose we had an ensemble of such estimators, possibly infinite, with $\mathbb{E}_T$ representing expectation with respect to this ensemble. Let $\ell$ be the KL-divergence loss function $\ell(z, f) = K(z \| f)$. Then, the (normalized) centroid of the distribution is given by*

$$\mathring{f}(y) = \frac{1}{Z} \exp(\mathbb{E}_T[\ln f(y)]), \tag{2}$$

*where $Z$ is a normalization constant independent of $y$.*

**Ensemble Combiner** A special example of the centroid is the centroid over ensemble members called the *ensemble combiner*, which we denote with the symbol $\bar{q}$. Let $T$ be the random variable distributed according to a discrete *model distribution*, such that each subpredictor $q_{(i)}$ can be drawn with probability $p(q_{(i)})$. Then, for the case of square loss, the combiner $\bar{q}$ can be found through Lemma 1 as

$$\bar{q}(x) = \sum_{i=1}^m p(q_{(i)}) q_{(i)}(x), \tag{3}$$

where we have replaced the expected value $\mathbb{E}_T$ by the weighted average $\sum_{i=1}^{m} p(q_{(i)})$. Intuitively, the weights $p(q_{(i)})$, which sum to unity, can represent our 'belief' in the output of subpredictor $q_{(i)}$ (Krogh & Vedelsby, 1994). In the case where $p(q_{(i)}) = 1/m$ for all $i \in [m] := \{1, ..., m\}$, we recover the simple average combiner from Section 3.1. Similarly, for the case of the KL-divergence, one finds the *logarithmic opinion pool* (Heskes, 1997; 1998) which is given through Lemma 2 as

$$\bar{q}(x) = \frac{1}{Z} \exp \left( \sum_{i=1}^{m} p(q_{(i)}) \ln q_{(i)}(x) \right) = \frac{1}{Z} \prod_{i=1}^{m} q_{(i)}(x)^{p(q_{(i)})}, \tag{4}$$

where we emphasize that each $q_{(i)}$ is a normalized probability distribution. As in Wood et al. (2023), we note that this is not necessarily the optimal way to combine the output of the subpredictors. However, for the bias-variance-diversity decomposition that we review in Section 3.3, the centroid from Definition 1 provides a framework that neatly allows us to decompose the expected risk.

### 3.3 BIAS-VARIANCE-DIVERSITY DECOMPOSITION

Up to now, we have worked with a general random variable $T$ and, for the model distribution, weights $p(q_{(i)})$. We now specialize to $T$ representing the training sets of size $n$ which we represent with the symbol $D$. We also specialize to uniform model weights $p(q_{(i)}) = 1/m$ for all $i \in [m]$. Finally, thanks to Lemma 3 (see Appendix A.2.2), one finds the main result from Wood et al. (2023) which we state here.

**Theorem 1** (Bias-variance-diversity decomposition). *Let $\{q_{(i)}\}_{i=1}^{m}$ be an ensemble of subpredictors and let $D$ be the random variable that represents training sets of size $n$. Let $\ell$ be a loss function that permits a valid bias-variance decomposition. Then, the expected value of the risk $R[\bar{q}] = \mathbb{E}_{XY}[\ell(y, \bar{q})]$ over all training sets $\mathcal{D}$ can be decomposed as*

$$\mathbb{E}_D \mathbb{E}_{XY}[\ell(y, \bar{q})] =$$

$$\mathbb{E}_X \Bigg[ \underbrace{\mathbb{E}_{Y|X}[\ell(y, y^*)]}_{noise} + \underbrace{\frac{1}{m} \sum_{i=1}^{m} \ell(y^*, \mathring{q}_{(i)})}_{average\ bias} + \underbrace{\frac{1}{m} \sum_{i=1}^{m} \mathbb{E}_D[\ell(\mathring{q}_{(i)}, q_{(i)})]}_{average\ variance} - \underbrace{\frac{1}{m} \sum_{i=1}^{m} \mathbb{E}_D[\ell(\bar{q}, q_{(i)})]}_{diversity} \Bigg],$$

*where $y^* = \mathbb{E}_{Y|X}[y]$, $\mathring{q}_{(i)} = \arg\min_z \mathbb{E}_D[\ell(z, q_{(i)})]$ is the centroid for a subpredictor, and $\bar{q} = \arg\min_z \sum_{i=1}^{m} p(q_{(i)})\ell(z, q_{(i)})$ is the ensemble combiner.*

In comparison to the decompositions in Geman et al. (1992) and Heskes (1998) for a single model, the decomposition for an ensemble can be decomposed into three terms, namely, a bias, variance, and diversity term (as well as an irreducible noise term). Importantly, observe that the diversity term comes with a negative sign. Thus, the greater the diversity, the lower the expected risk. We refer the reader to Wood et al. (2023) for further theoretical results and experimental verification of Theorem 1.

## 4 IMPLICIT ENSEMBLES AND IMPLICIT REGULARIZATION

Having established the theoretical framework for ensembles in the previous section, we now turn to the case of a *single* feedforward neural network with ReLU activations. We extend the theoretical framework for ensembles by viewing the neural network as an *implicit ensemble*.

### 4.1 SETUP

We consider any neural network with fully connected (FC) layers at the end with ReLU activations. More precisely, let $f = h \circ g : \mathcal{X} \subset \mathbb{R}^{d_i} \to \mathcal{Y} \subset \mathbb{R}^{d_f}$ represent a neural network, where $g : \mathcal{X} \subset \mathbb{R}^{d_i} \to \mathbb{R}^{d_0}$ is any flattened feature map (such as a set of convolutional layers) and $h : \mathbb{R}^{d_0} \to \mathcal{Y} \subset \mathbb{R}^{d_f}$ is $L$-hidden FC layers, with hidden widths $d_1, ..., d_L \in \mathbb{N}$, that uses ReLU activations $\sigma(x) = \max(0, x)$. For $h$, we consider a $\{\alpha, \beta, H\}$-family of FC layers as follows:

$$\begin{aligned}
h_{(1)}(x) &= \sigma(z_{(1)}(x)), \quad z_{(1)}(x) = \alpha w_{(1)} h_{(0)}(x), \\
h_{(\ell)}(x) &= \sigma(z_{(\ell)}(x)), \quad z_{(\ell)}(x) = w_{(\ell)} h_{(\ell-1)}(x), \quad \ell > 1, \\
h_{(L+1)}(x) &= \beta w_{(L+1)} h_{(L)}(x),
\end{aligned} \tag{5}$$

where $h_{(0)}(x) = g(x) \in \mathbb{R}^{d_0}$ is the input to the FC layers, $z_{(\ell)}(x) \in \mathbb{R}^{d_\ell}$ is the preactivation vector to layer $\ell$, $h_{(\ell)}(x) \in \mathbb{R}^{d_\ell}$ is the activation vector of layer $\ell$, $h_{(L+1)}(x) \in \mathbb{R}^{d_f}$ is the output of the neural network, and $w_{(\ell)} \in \mathbb{R}^{d_\ell \times d_{\ell-1}}$ is a learnable weight matrix with elements initialized as $w^i_{(\ell)\ j} \overset{iid}{\sim} H$. The family of neural networks are determined by the (non-learnable) parameters $\alpha, \beta \in \mathbb{R}$ and the distribution $H$ and, as we will discuss later, plays a nontrivial role in the infinite width limit of the FC layers (Yang & Hu, 2022). Important to our case, the parameter $\beta$ will lead to different identifications of subpredictors within the neural network. We consider three particular choices of $\{\alpha, \beta, H\}$ called, namely, *standard parameterization* (SP) (Paszke et al., 2019), *mean-field parameterization* (MFP) (Mei et al., 2018) and *maximal-update parameterization* ($\mu P$) (Yang & Hu, 2022) – their values are listed in Table 1.

Table 1: Parameterizations for FC layer

| Parameterization | L | $\alpha$ | $\beta$ | H |
|---|---|---|---|---|
| Standard | $\geq 1$ | 1 | 1 | $\text{Uniform}(-\sqrt{\frac{6}{d_{\ell-1}}}, \sqrt{\frac{6}{d_{\ell-1}}})$ |
| Mean-field | 1 | 1 | $\frac{1}{d_L}$ | $\mathcal{N}(0, 1)$ |
| Maximal-update | $\geq 1$ | $\sqrt{d_0}$ | $\frac{1}{\sqrt{d_L}}$ | $\mathcal{N}(0, \frac{1}{d_{\ell-1}})$ |

Finally, for the case of a $C$-class classification task, we use a Softmax function on the output

$$\text{Softmax}[h_{(L+1)}(x)] = \frac{1}{Z} \exp(h_{(L+1)}(x)), \quad Z = \sum_{c=1}^{C} \exp(h^c_{(L+1)}(x)), \tag{6}$$

where the exponent is applied component-wise.

### 4.2 SUBPREDICTOR IDENTIFICATION

**Regression** We first consider a regression task with square loss (see Figure 2). We apply the theoretical framework of Section 3 by working backwards from the definition of the combiner to formulate a notion of subpredictor within the neural network. The combiner $\bar{q}$ in this case is fixed – it is, of course, the output nodes of the neural network $h_{(L+1)}(x)$.

Now, consider Equation 3. We need to identify both subpredictors $q_{(i)}$ and model weights $p(q_{(i)})$, where we are *constrained* to $\sum_{i=1}^{m} p(q_{(i)}) = 1$ and $p(q_{(i)}) \geq 0$ for all $i \in [m]$. The constraint limits our choices of possible subpredictors. Setting $\bar{q} = h_{(L+1)}$, we have at the level of components

$$\bar{q}^i(x) = \beta w^i_{(L+1)} h_{(L)}(x) = \beta \sum_{j=1}^{d_L} w^i_{(L+1)\ j} h^j_{(L)}(x), \tag{7}$$

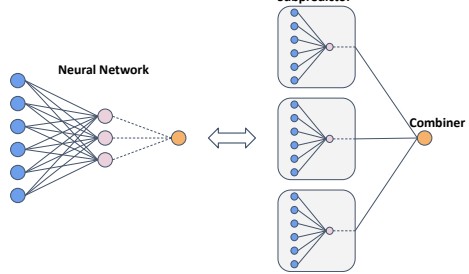

Figure 2: **Implicit ensemble** A single hidden layer neural network can be viewed as an ensemble. Each subpredictor of the ensemble consists of a hidden node multiplied by a weight. The subpredictor outputs are combined with weights $p(q_{(i)})$ to form the combiner.

where $w^i_{(L+1)} \in \mathbb{R}^{1 \times d_L}$ (i.e. the $i$th row of $w_{(L+1)}$). By comparing to Equation 3, we observe that we can naturally satisfy the constraint if $\sum_{i=1}^{d_L} \beta = 1$ and $\beta$ is nonnegative, which is satisfied if $\beta = 1/d_L$. Interestingly, this is naturally realized by MFP for $L = 1$ (i.e. a single hidden layer MLP). Consequently, in this case, a subpredictor is naturally identified as $q_{(j)}(x) = w_{(2)\ j} h^j_{(1)}(x) \in \mathbb{R}^{d_f}$, $j \in [d_1]$ (where $w_{(2)\ j} \in \mathbb{R}^{d_f}$, i.e. the $j$th column of $w_{(2)}$).

For SP and $\mu P$, we can still satisfy the constraint and identify valid subpredictors. For SP, $\beta = 1$, hence, we can factor out $1 = 1/d_L \times d_L$ so that $p(q_{(i)}) = 1/d_L$ and $q_{(j)}(x) = d_L w_{(L+1)\ j} h^j_{(L)}(x)$. Similarly, for $\mu P$, $\beta = 1/\sqrt{d_L}$, and so we can factor $1/\sqrt{d_L} = 1/d_L \times \sqrt{d_L}$, giving weight

$p(q_{(i)}) = 1/d_L$ and subpredictor $q_{(j)}(x) = \sqrt{d_L} w_{(L+1)\ j} h^j_{(L)}(x)$. We therefore find that, unlike the MFP case, the subpredictors for SP and $\mu P$ comes with a scalar factor determined by the width of the last hidden layer.

Furthermore, using Lemma 1, we can immediately identify the centroid for each subpredictor as $\mathring{q}_{(i)}(x) = \mathbb{E}_D[q_{(i)}(x)]$. For example, for the MFP case, $\mathring{q}_{(i)}(x) = \mathbb{E}_D[w_{(2)\ i} h^i_{(1)}(x)]$, where we note that both the weights $w_{(2)}$ and the hidden nodes $h_{(1)}$ depend on the training set $\mathcal{D}$. Results are summarized in Table 2 for the MFP (the other parameterizations are similar).

**Classification** For classification, the neural network output is given by Equation 6 where it now outputs a *probability vector* (components are nonnegative and sum to one). Recall that, for the KL-divergence, the combiner is given by Equation 4. For the case of classification, the identification of subpredictors is more nuanced. As before, we have the constraint that the model distribution weights $p(q_{(i)})$ need to be nonnegative and sum to one. Additionally, we now require each subpredictor to output a valid probability vector: $q_{(i)}(x) \in \mathbb{R}^C$, $\sum_{c=1}^C q^c_{(i)}(x) = 1$, $q^c_{(i)}(x) \geq 0$. Using the Softmax function from Equation 6, we identify the subpredictors as in Table 2 for the MFP (the other parameterizations are similar). See Appendix A.3.1 for details.

Table 2: Framework components for 1-layer neural networks initialised with MFP.

| | square loss | | KL-loss | |
|---|---|---|---|---|
| subpredictor | $q_{(i)}(x) = w_{(2)\ i} h^i_{(1)}(x)$ | | $q_{(i)}(x) = \text{Softmax}(w_{(2)\ i} h^i_{(1)}(x))$ | |
| centroid | $\mathring{q}_{(i)}(x) = \mathbb{E}_D[w_{(2)\ i} h^i_{(1)}(x)]$ | (8) | $\mathring{q}_{(i)}(x) = \text{Softmax}(\mathbb{E}_D[\ln q_{(i)}(x)])$ | (9) |
| combiner | $\bar{q}(x) = \frac{1}{d_1} \sum_{i=1}^{d_1} q_{(i)}(x)$ | | $\bar{q}(x) = \text{Softmax}\left( \frac{1}{d_1} \sum_{i=1}^{d_1} \ln q_{(i)}(x) \right)$ | |

**Discussion** For two popular loss functions, the square-loss and KL-loss, we have shown that a single neural network can be reinterpreted as an implicit ensemble. In particular, we have shown that the notion of a subpredictor within the neural network is dependent on the choice of loss function used for training. Naively, one might associate a hidden node to a subpredictor but, instead, we find that a subpredictor typically consists of a hidden node multiplied by an outgoing weight vector. Apart from its contribution to the output of the subpredictor, the role of the outgoing weight vector $w_{(L+1)\ j} \in \mathbb{R}^{d_f}$ has additional importance: it carries the necessary index structure to ensure that the subpredictor is a vector (if the neural network output is a vector). For example, for a subpredictor $j$, $q^i_{(j)} = w^i_{(L+1)\ j} h^j_{(L)}$ carries a free index $i$ (due to the weight vector) that ensures that $q_{(j)}$ is a vector. This substructure in the neural network naturally emerges when we apply the framework of Section 3. Finally, we also see that the number of subpredictors within a single neural network is determined by the width $d_L$ of the last hidden layer.

In order to perform these identifications, we have considered three parameterizations of the FC layers. Notably, for MFP, this choice of parameterization already includes the correct factor $p(q_{(i)}) = 1/d_1$ on the output nodes; in fact, this factor is also needed to guarantee feature learning in the large width limit (Chizat et al., 2019; Mei et al., 2018) [1]. Unlike MFP, SP and $\mu P$ carry an explicit factor dependent on $d_L$ (the last hidden layer width) in the definition of the subpredictor and, thus, risk diverging in the large width limit. Although beyond the scope of this paper, we consider the potential connection between the implicit ensemble view developed here and the feature learning/kernel regimes to be an interesting avenue of future work.

### 4.3 DECOMPOSITION

Using the identifications in the previous section, we can now apply Theorem 1 to the case of a single neural network. For regression using least squares loss, we assume a training set $\mathcal{D} = $

---

[1]Both MFP and $\mu P$ belong to a family of abc-parameterizations in the rich regime of feature learning (note that MFP is a special case of $\mu P$ modulo a symmetry transformation) (Yang & Hu, 2022). In contrast to the rich regime, standard parameterization (SP) belongs to the kernel or lazy regime (Jacot et al., 2018; Chizat et al., 2019; Yang & Hu, 2022).

$\{(x^{(i)}, y^{(i)})\}_{i=1}^n$ where $y^{(i)} \in \mathbb{R}$. With the identification in Equation 8, we find that

$$\mathbb{E}_D \mathbb{E}_{XY}[(y - \bar{q})^2] =$$

$$\mathbb{E}_X \left[ \underbrace{\mathbb{E}_{Y|X}[(y - y^*)^2]}_{\text{noise}} + \underbrace{\frac{1}{d_L} \sum_{i=1}^{d_L} (y^* - \mathring{q}_{(i)})^2}_{\text{average bias}} + \underbrace{\frac{1}{d_L} \sum_{i=1}^{d_L} \mathbb{E}_D[(\mathring{q}_{(i)} - q_{(i)})^2]}_{\text{average variance}} - \underbrace{\frac{1}{d_L} \sum_{i=1}^{d_L} \mathbb{E}_D[(\bar{q} - q_{(i)})^2]}_{\text{diversity}} \right],$$

(10)

where $y^* = E_{Y|X}[y]$.

For classification, we assume a training set $\mathcal{D} = \{(x^{(i)}, , y^{(i)})\}_{i=1}^n$ where $y^{(i)} \in \mathbb{R}^C$ is a one-hot vector. Then, the KL-divergence reduces to[2]

$$K(y||\bar{q}) = -y \cdot \ln \bar{q} = \text{cross-entropy}(y, \bar{q}).$$

(11)

Using the identifications in Equation 9, we find that

$$\mathbb{E}_D \mathbb{E}_{XY}[\text{cross-entropy}(y, \bar{q})] =$$

$$\mathbb{E}_X \left[ \underbrace{\mathbb{E}_{Y|X}[K(y||y^*)]}_{\text{noise}} + \underbrace{\frac{1}{d_L} \sum_{i=1}^{d_L} K(y^*||\mathring{q}_i)}_{\text{average bias}} + \underbrace{\frac{1}{d_L} \sum_{i=1}^{d_L} \mathbb{E}_D[K(\mathring{q}_i||q_i)]}_{\text{average variance}} - \underbrace{\frac{1}{d_L} \sum_{i=1}^{d_L} \mathbb{E}_D[K(\bar{q}||q_i)]}_{\text{diversity}} \right],$$

(12)

where $y^* = \mathbb{E}_{Y|X}[y]$ is the true class distribution at $x$ (see Appendix B.3 of Wood et al. (2023)). Note that the noise reduces to $\mathbb{E}_{Y|X}[K(y||y^*)] = -y^* \cdot \ln y^*$, which is the Shannon entropy of the true class distribution at $x$.

Similar to the case for explicit ensembles, the exposed diversity term for a single neural network comes with a negative sign. Since the risk is always positive, the diversity term must be less than or equal to the sum of the bias and variance terms. Due to the sign and the fact that the loss functions are positive, the diversity term reduces the error introduced through the bias and variance terms. Consequently, we identify this as a new source of *implicit regularization* that helps neural network generalization, and that was hidden in the original decomposition of Geman et al. (1992) for neural networks.

### 4.4 DISCUSSION

By viewing a neural network as an implicit ensemble, we gain insight into their implicit regularization.

**Overfitting**    For the case of high variance error, the diversity term acts as a regularizer that helps prevent the neural network from overfitting its training set (for sufficiently large diversity). For the case of wide overparameterized models that interpolate their training data yet still generalizes well, we hypothesize that the implicit regularization is partly due to the hidden diversity derived from the subpredictors of the neural network. We empirically validate our hypothesis in Section 5.

**Shallow Network Capacity**    High bias error is attributed to a low capacity model that is unable to fit their training data well. A neural network is constructed out of simpler parts. In particular, as we have argued using the implicit ensemble picture, the neural network is constructed out of smaller submodels or subpredictors. For shallow networks like in Figure 2, these subpredictors are relatively simple models (a hidden node multiplied by a weight) and each subpredictor will likely have a high bias error. However, once combined into a neural network, the bias error of the subpredictors will be reduced due to the effect of their diversity. Thus, by combining low capacity parts (or models), a neural network is able to reduce the high bias error thanks to the effects of diversity. We demonstrate this in Section 5.

## 5 EXPERIMENTS: ESTIMATING BIAS, VARIANCE, AND DIVERSITY

In this section, we empirically verify our decompositions from Section 4 by estimating the bias, variance, and diversity for a large number of trained neural networks. This allows us to explore how

---

[2]We use the fact that $\lim_{x \to 0} x \ln x = 0$.

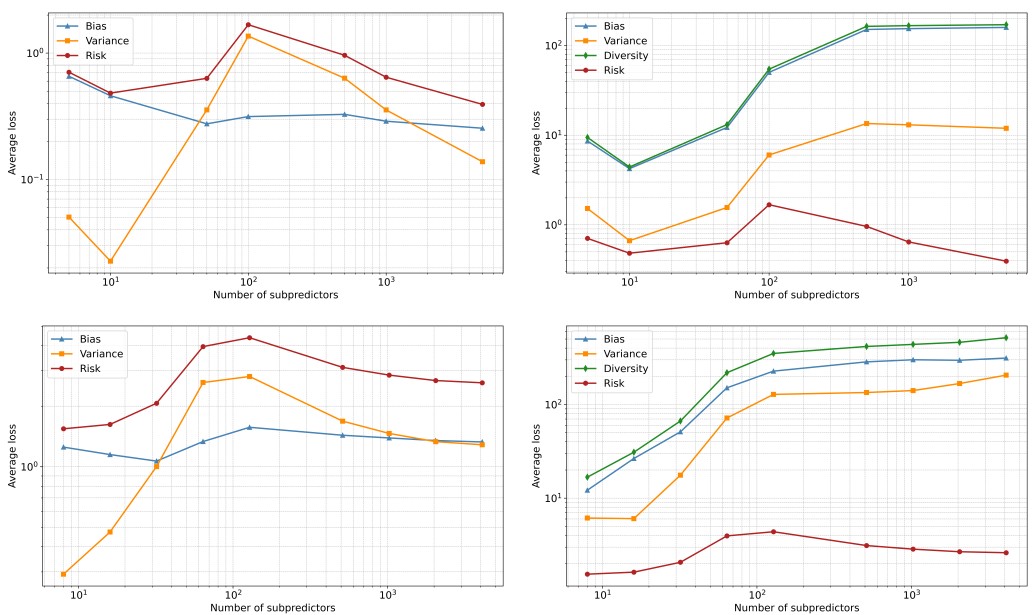

Figure 3: The estimated bias-variance decomposition (left) and bias-variance-diversity decomposition (right) for models with an increasing number of subpredictors trained on MNIST (top) and CIFAR10 (bottom). The estimates are calculated over 50 trials for each model, where each trial is trained on a randomly sampled subset of the train set. Note that the risk curves are identical between left and right (only the y-axis scale differs).

these measurements vary as the number of subpredictors and the corresponding risk varies, and how this compares to the traditional bias-variance decomposition. Here we demonstrate this on the specific case of double descent (DD) (Belkin et al., 2019; Nakkiran et al., 2020) as it provides an "interesting" risk curve with varied behavior, exhibiting three regimes, namely, the underparameterized regime, the critical regime, and the overparameterized regime. We then demonstrate the regularization effect of diversity on the variance error.

To show the wide applicability and utility of our framework we consider two very different settings: 1) Shallow one layer MLPs trained on MNIST, and 2) Deep three layer CNNs trained on CIFAR10. We then also confirm our observations on many additional datasets and training setups in Appendix C.

**Setup** For both MNIST and CIFAR10, we train models of increasing capacity with additional label noise in order to elicit a DD. For each model capacity, we repeat this over 50 samplings of the dataset (trials) [3], where we average over these trials in order to estimate the expectation over training datasets ($\mathbb{E}_D$ in Equation 12). This results in approximately 450 models that need to be trained for each decomposition – see Appendix B.1 for details. Additionally, we also use these trials to estimate the traditional bias and variance decomposition for each model Geman et al. (1992).

For **MNIST**, we consider one hidden layer fully-connected feedforward networks with the mean-field parameterization trained with cross-entropy loss on a randomly sampled $90\%$ of the training data per trial. To minimize some of the implicit regularization effects due to mini-batch SGD (Smith et al., 2021), we make use of full batch gradient descent as an optimizer. For **CIFAR10**, we train three layer CNNs (two convolutional, one fully connected layers) with the standard parameterization, cross-entropy loss, and mini-batch SGD on a randomly sampled $10\%$ of the training data per trial. These two very different settings allow use to illustrate the wide applicability of our theoretical framework. Further hyperparameter details for these models can be found in Appendix B.2.

---

[3]Note that the random initialization seed for each model is fixed over trials, only the randomly sampled dataset differs.

**Implicit regularization** The results of our analysis are shown in Figure 3 for both MNIST (top) and CIFAR10 (bottom), including the traditional bias-variance decomposition (left) and the bias-variance-diversity decomposition (right). For both MNIST and CIFAR10 we observe a clear DD in the risk curves. When considering classic decomposition (left) we observe that these changes in risk are predominately driven by changes in the variance term, which increases rapidly up to a maximum at the critically-parameterized regime (approximately $10^2$ subpredictors for both MNIST and CIFAR10), before it *inexplicably decreases as the number of subpredictors is further increased.* However, when considering the bias-variance-diversity curve (right) we observe a remarkably different view of this behavior. Specifically, we note that all three terms (bias, variance, diversity) *increase*, and appear to keep increasing past the critically-parameterized regime. It is also evident that the bias and diversity terms closely track each other, with the diversity term slightly larger, although this difference is difficult to see on the logarithmic scale.

In order to better understand the effect of diversity on the variance error, we plot the difference between the bias and diversity, which we refer to as the *diversity residual*, along with the variance and risk on a linear scale. In this case, since the diversity is larger than the bias, the diversity residual is negative, therefore we plot the absolute value. In this formulation, the risk is given by risk = variance − |diversity residual|. The size of the diversity residual term therefore directly indicates the size of the regularization effect which stems from the diversity. See Figure 1.

For both the MNIST models and CIFAR10 models, we observe that the diversity residual follows the same shape as that of the variance error. Initially, in the underparameterized regime ($< 10^2$ subpredictors), the variance error grows faster than the diversity residual which results in an increasing risk. Remarkably, after crossing the interpolation point (approximately 100 subpredictors) into the overparameterized regime, the diversity residual grows more rapidly than the variance error (despite the variance error still increasing) which directly results in a decreasing risk (i.e. the second descent). Put simply, we observe that the subpredictors become more 'wrong' individually (bias and variance grows) but they also become dramatically more diverse, and the diversity grows faster than the bias and variance. **This increased diversity explains the drop in test risk that is otherwise hidden in the classical bias-variance decomposition.**

These results validate our main hypothesis that the diversity term implicitly regularizes the variance error and, for networks in the overparameterized regime, helps mitigate the effects of overfitting (see Section 4.4). Additionally, we also observe in Figure 3 that the subpredictors contribute a high bias error, as anticipated in Section 4.4, but that this error is mitigated by the larger diversity term.

**Additional evidence** To further confirm our results we repeat similar experiments on other datasets such as MNIST without label noise, MNIST with less data, MNIST with mini-batch training, Fashion MNIST (Xiao et al., 2017), and a regression task. Additionally, we also consider two layer MLPs and other parameterization. In all cases, we find similar results and behaviors as discussed here, namely that 1) the diversity terms closely tracks the bias, and importantly 2) the diversity residual term grows faster than the variance term as capacity is increased in the overparameterized regime. See Appendix C for these additional experiments.

## 6 CONCLUSION

By extending the theoretical framework of Section 3 to interpret a neural network as an implicit ensemble, we found a new bias-variance decomposition that includes a diversity term (Section 4). This additional term, which comes with a negative sign, acts as an implicit regularizer. We demonstrated this implicit regularization for several different datasets, architectures and network parameterizations for two losses, namely, the square loss and the cross-entropy loss. For the case of double descent, we found that, contrary to the original bias-variance decomposition, both the bias and the variance continues to increase after the interpolation point. However, as can be clearly seen when subtracting the diversity from the bias, the residual diversity both tracks and increases more rapidly than the variance, thus regularizing the variance error in the overparameterized regime. This phenomenon has not previously been observed and provides a new perspective on the ability of neural networks to self-regularize.

**Future work** While this work has demonstrated a new implicit regularizer in neural networks, there are interesting open questions that remain. Firstly, as seen in Figure 1 (as well as in the additional experimental results in Appendix C), the reason as to why diversity closely tracks the variance is currently being investigated. Additionally, the diversity is seen to increase more rapidly when transitioning to the overparameterized regime. We consider the reason behind this change in the behavior of the diversity term a highly interesting avenue of further research. We also consider the extension of our framework to architectures such as transformers to be an important next step. Finally, we believe that the work in this paper can lead to novel designs for training algorithms that enhances the diversity of the subpredictors. For existing methods such as Dropout (Srivastava et al., 2014), we conjecture that it enhances the diversity term in the neural network's bias-variance-diversity decomposition.

More tangential to this paper, we remarked in Section 4.2 that the MFP is already in the correct form for our construction of the implicit ensemble. More generally, the implicit ensemble picture developed here for various parameterizations and their connection with the feature-learning regimes or kernel regimes may be an interesting avenue of research. We also mention the study of Golubeva et al. (2021) that showed that width plays a more primary role in generalization when compared to number of parameters. We hypothesize that this is due to the effect of increasing diversity and believe that this should be testable using the methods of Golubeva et al. (2021). Finally, while we did investigate the effects of mini-batch versus full batch training (Appendix C.1), we also consider the general effects of the training algorithm on the diversity of the subpredictors to be potentially interesting future work.

## 7 REPRODUCIBILITY STATEMENT

For Section 3, we provide proofs and extra details in Appendix A.2. For Section 4.2, we provide additional details in Appendix A.3 for the classification case. For our experiments, we provide experimental details in Appendix B; in particular, this appendix includes the method we used for estimating bias, variance, and diversity (Appendix B.1), the estimation equations for the terms in the decomposition (Appendix B.1.1 and Appendix B.1.2), and a full description of our experimental details and hyperparameter settings (Appendix B.2). Additionally, we also include our code repository as supplementary material for further reproducibility of our results.

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

# A ADDITIONAL THEORY

In this appendix, we provide additional theoretical details.

The appendix is organized as follows: in Appendix A.1, we discuss the limitations of our theoretical method. In Appendix A.2, we provide proofs in support of Section 3. In Appendix A.3, we provide supplementary details concerning the derivation of the subpredictors in the case of the KL-divergence.

## A.1 LIMITATIONS

In this section, we discuss the limitations of our approach.

**Loss functions**  Our analysis is valid for any loss function that satisfies Definition 2. This can, in principle, be applied to any loss function derived from a *Bregman divergence* (Wood et al., 2023). In principle, the decomposition can also be extended to any loss function by distinguishing between the measurement of bias and variance, and the measurement of its effect, as defined by James and Hastie (James & Hastie, 1997) and demonstrated in (Wood et al., 2023). In this paper, however, we only analyze squared error and cross-entropy loss functions, and extensions will be required to handle loss functions such as 0/1-loss and absolute loss.

**Architectures**  Our current theoretical framework is limited to any architecture that has a $L$-hidden layer feedforward ReLU network at the end. As this is the case for many types of architectures, our framework covers a wide range of use cases.

## A.2 PROOFS

In this section, we provide proofs and extra details around the background section of our paper (Section 3). For more details on the background section, see the work of Wood et al. (2023).

### A.2.1 PROOF OF LEMMA 1 AND LEMMA 2

We start with the proof for Lemma 1:

*Proof.* By Definition 1, we need to compute $\mathring{t} = \arg\min_z \mathbb{E}_T[(t - z)^2]$. We note that since $z \in \mathcal{B}$ is a nonrandom number,

$$
\begin{aligned}
0 &= \frac{d}{dz}\mathbb{E}_T[(t - z)^2] \\
&= -2\mathbb{E}_T[(t - z)] \\
&\Rightarrow z = \mathbb{E}_T[t].
\end{aligned}
$$

$\square$

Next, we provide the proof of Lemma 2:

*Proof.* By Definition 1, we need to compute $\mathring{p} = \arg\min_z \mathbb{E}_T[K(z||p)]$. Unlike the least squares loss, $z$ is now a function and we therefore need to use a functional derivative to compute

$$
\arg\min_z \mathbb{E}_T[K(z||p)] \quad \text{s.t.} \quad \int dy\, z(y) = 1.
$$

We use the following Lagrangian with a Lagrange multiplier $\lambda$ to enforce the constraint:

$$
\mathcal{L}[z(y)] = \int dy\, z(y)\ln z(y) - z(y)\mathbb{E}_T[\ln p(y)] + \lambda(z(y) - 1).
$$

Then, for an arbitrary variation $z(y) \mapsto z(y) + \epsilon\delta\phi(y)$,

$$
\begin{aligned}
\frac{d}{d\epsilon}\mathcal{L}[z(y) + \epsilon\delta\phi(y)]\Big|_{\epsilon=0} &= \int dy\, \big(\ln z(y) - \mathbb{E}_T[\ln p(y)] + \lambda\big)\delta\phi(y) = 0 \\
&\Rightarrow z(y) = \frac{1}{Z}\exp(\mathbb{E}_T[\ln p(y)]),
\end{aligned}
$$

where $Z$ is a normalization constant independent of $y$. $\square$

The above result can be easily extended to the discrete case.

### A.2.2 PROOF OF THEOREM 1

Using the notion of a centroid, Wood et al. (2023) gave a generalized definition of the bias-variance decomposition, which we state here.

**Definition 2** (Generalized bias-variance decomposition). *Let $T$ be a random variable and let $\ell$ be a loss function. Let $R[q] = \mathbb{E}_{XY}[\ell(y, q(x))]$ be the risk for a model that depends on $T$, $q = q(x; t)$, $T = t \in \mathcal{B}$. If the following form holds, then we refer to it as a* generalized bias-variance decomposition

$$\mathbb{E}_T R[q] = \mathbb{E}_X \left[ \underbrace{\mathbb{E}_{Y|X}[\ell(y, y^*)]}_{noise} + \underbrace{\ell(y^*, \mathring{q})}_{bias} + \underbrace{\mathbb{E}_T[\ell(\mathring{q}, q)]}_{variance} \right], \tag{13}$$

*where $y^* = \mathbb{E}_{Y|X}[y]$ and $\mathring{q}$ is as defined in Definition 1.*

For example, if the random variable $T$ represents the training datasets $\mathcal{D}$ of size $n$ drawn from $P(X, Y)^n$ and the loss function $\ell$ is the square loss, then we recover the bias-variance decomposition from Geman et al. (1992) using Lemma 1. Similarly, if $\ell$ is the KL-divergence, then we recover the analogous decomposition from Heskes (1998) using Lemma 2.

We next introduce a special term called the *ambiguity*, which was first derived in Krogh & Vedelsby (1994), and which can be interpreted as the variance of the weighted ensemble around the weighted average. Intuitively, given an input $x$, it measures the degree of disagreement among the subpredictors of the ensemble.

**Definition 3** (Ambiguity). *Let $\{q_{(i)}\}_{i=1}^m$ be an ensemble of subpredictors and let $\bar{q} = \arg\min_z \sum_{i=1}^m p(q_{(i)})\ell(z, q_{(i)})$ be their ensemble combiner. Then, given an input $x$, the* ensemble ambiguity $a(x)$ *over the model distribution is defined as*

$$a(x) = \sum_{i=1}^m p(q_{(i)})\ell(\bar{q}(x), q_{(i)}(x)).$$

A consequence of this definition is that one can decompose the loss $\ell(y, \bar{q})$ between the combiner output $\bar{q}$ and the target $y$ into a term involving the weighted average loss (between the subpredictors and $y$) and the ambiguity (Krogh & Vedelsby, 1994; Wood et al., 2023).

**Lemma 3** (Generalized ambiguity decomposition). *Let $\ell$ be a loss function that allows a bias-variance decomposition in the sense of Definition 2. Then, for a pair $(x, y)$ and for an ensemble of subpredictors $\{q_{(i)}\}_{i=1}^m$, the* generalized ambiguity decomposition *is given by*

$$\ell(y, \bar{q}) = \sum_{i=1}^m p(q_{(i)})\ell(y, q_{(i)}) - \sum_{i=1}^m p(q_{(i)})\ell(\bar{q}, q_{(i)}).$$

*Proof.* By assumption, $\ell$ permits a decomposition in the sense of Definition 2. Let the random variable $T$ represent a subpredictor $T = q_{(i)}$ drawn from a discrete model distribution with probability $p(q_{(i)})$. Then, replacing the expectation value $\mathbb{E}_T$ with an average $\sum_{i=1}^m p(q_{(i)})$ and using the combiner $\bar{q} = \arg\min_{z \in \mathcal{B}} \sum_{i=1}^m p(q_{(i)})\ell(q_{(i)}, z)$ for $\mathring{q}$, Definition 2 gives

$$\mathbb{E}_{XY}\left[ \sum_{i=1}^m p(q_{(i)})\ell(q_{(i)}, y) \right] = \mathbb{E}_{XY}\left[ \ell(y, \bar{q}) + \sum_{i=1}^m p(q_{(i)})\ell(\bar{q}, q) \right],$$

where we assumed, without loss of generality, that the noise term is zero [4]. Equating the terms in the parentheses and rearranging gives the required result. □

We can now prove Theorem 1 as follows:

---

[4] If a noise term is present, then one can show that $\mathbb{E}_{Y|X}[\ell(y, y^*)] + \ell(y^*, q) = \mathbb{E}_{Y|X}[\ell(y, q)]$ for a valid loss function. For example, see the original derivation in (Geman et al., 1992) for square loss and the derivation in (Heskes, 1998) for KL-divergence.

*Proof.* From Lemma 3, we find that

$$\mathbb{E}_D \mathbb{E}_{XY}[\ell(y, \bar{q})] = \frac{1}{m} \sum_{i=1}^m \mathbb{E}_D \mathbb{E}_{XY}[\ell(y, q_{(i)})] - \frac{1}{m} \sum_{i=1}^m \mathbb{E}_D \mathbb{E}_{XY}[\ell(\bar{q}, q_{(i)})].$$

Applying Definition 2 to the first term yields the required result. □

### A.3 CLASSIFICATION SUPPLEMENTARY

By extending the framework of Section 3 to a single neural network, we derived subpredictors for two loss function in Section 4. In this section, we provide additional details for the case of the Kullback-Leibler divergence loss. In Section A.3.1, we explicitly derive the subpredictors in the neural network for the KL-loss.

#### A.3.1 SUBPREDICTORS DERIVATION

Similar to the regression case, the combiner $\bar{q}$ is already fixed and is given by Equation 6, which we write in component form as

$$\bar{q}^c(x) = \frac{1}{Z(x)} \exp\left( \beta \sum_{i=1}^{d_L} w_{(L+1)\,i}^c h_{(L)}^i(x) \right), \quad Z(x) = \sum_{c=1}^{C} \exp\left( \beta \sum_{i=1}^{d_L} w_{(L+1)\,i}^c h_{(L)}^i(x) \right). \tag{14}$$

We follow the same approach used for regression and work backwards to identify the subpredictors in the implicit ensemble. For the discrete model distribution weights $p(q_{(i)})$, recall that we have the constraints that the weights should sum to unity and be nonnegative. Additionally, we have the constraint that our subpredictors need to be normalized and output a valid probability vector (components need to be nonnegative and sum to unity). In comparison to Equation 4 and using our constraints, we naturally interpret the *normalized* subpredictor and combiner weights for the MFP case as (recall that $L = 1$ for MFP – see Table 1)

$$p(q_{(i)}) = \beta = \frac{1}{d_1}, \quad q_{(i)}^c(y) = \frac{1}{Z_i(x)} \exp(w_{(2)\,i}^c h_{(1)}^i(x)), \quad i \in [d_1], \quad c \in [C], \tag{15}$$

where $Z_i(x) = \sum_{c=1}^C \exp(w_{(2)\,i}^c h_{(1)}^i(x))$. Due to the normalization $Z_i(x)$ needed for each subpredictor, we therefore find that the combiner factorizes with an additional term as follows:

$$\bar{q}^c(x) = \frac{1}{Z} \exp\left( \frac{1}{d_1} \sum_{i=1}^{d_1} w_{2\,i}^c h_{(1)}^i(x) \right) = \frac{1}{Z} \exp\left( \frac{1}{d_1} \sum_{i=1}^{d_1} \ln q_i^c \right) \exp\left( \frac{1}{d_1} \sum_{j=1}^{d_1} \ln Z_j \right). \tag{16}$$

At first glance, our choice of subpredictor does not factor into the correct form seen in Equation 4. However, note that we can factor the combiner normalization $Z$ as follows:

$$Z = \sum_{c=1}^C \exp\left( \frac{1}{d_1} \sum_{j=1}^{d_1} w_{(2)\,j}^c h_{(1)}^j(x) \right) = \exp\left( \frac{1}{d_1} \sum_{j=1}^{d_1} \ln Z_j \right) \sum_{c=1}^C \exp\left( \frac{1}{d_1} \sum_{i=1}^{d_1} \ln q_{(i)}^c \right), \tag{17}$$

which yields the same term. Hence, the extra term can be canceled and the combiner $\bar{q}$ remains unchanged, as it should, while satisfying the form in Equation 4. Finally, the normalized centroid $\mathring{q}_{(i)}$ is given by Lemma 2

$$\mathring{q}_{(i)}(x) = \frac{1}{\mathring{Z}_j(x)} \exp(\mathbb{E}_D[\ln q_{(i)}(x)]), \tag{18}$$

where $\mathring{Z}_j(x) = \sum_{c=1}^C \exp(\mathbb{E}_D[\ln q_{(j)}^c(x)])$. Together, these identifications yield the results in Table 2 for the MFP case.

The same procedure also yields the subpredictors for the SP and $\mu P$ cases for $L \geq 1$. Similar to the regression case, their subpredictors are scaled by a factor determined by the last layer's hidden width $d_L$. For SP, we find that

$$p(q_{(i)}) = \frac{1}{d_L}, \quad q_{(i)}^c(y) = \frac{1}{Z_i(x)} \exp(d_L w_{(L+1)\,i}^c h_{(L)}^i(x)), \quad i \in [d_L], \quad c \in [C], \tag{19}$$

and for $\mu P$ we find that

$$p(q_{(i)}) = \frac{1}{d_L}, \quad q_{(i)}^c(y) = \frac{1}{Z_i(x)} \exp(\sqrt{d_L} w_{(L+1)\,i}^c h_{(L)}^i(x)), \quad i \in [d_L], \quad c \in [C]. \tag{20}$$

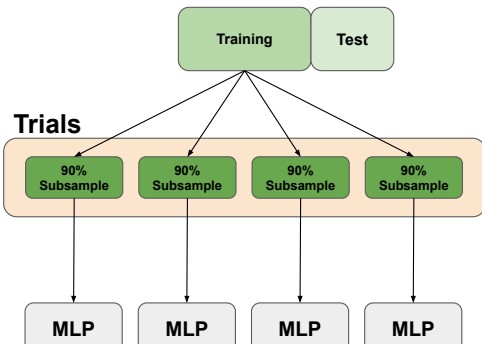

Figure 4: **Estimating bias, variance, diversity** We follow the example of Wood et al. (2023): after performing a train-test split, we randomly subsample $t$ trial sets from the training set. On each trial set, we train a model with fixed hyperparameters. We then estimate bias, variance, and diversity over the test set.

## B EXPERIMENTAL DETAILS

### B.1 ESTIMATING BIAS, VARIANCE, AND DIVERSITY

To estimate the bias, variance, and diversity terms given in Equation 10 and Equation 12, we follow the approach used in Wood et al. (2023) (see also Neal et al. (2019)). Below we provide a high-level overview of the method. In the next sections, we provide the precise equations to estimate the various terms.

There are two expectation values that we need to estimate, namely, $\mathbb{E}_{XY}$ and $\mathbb{E}_D$, where the random variable $D$ represents training sets of size $n$. Following Wood et al. (2023), the procedure is as follows (see Figure 4):

**Estimating $\mathbb{E}_{XY}$** We perform a standard train-test split of the data. We then use the test set to estimate $\mathbb{E}_{XY}$ by taking an average over the test samples. Let $s$ be the number of test sample pairs in our test set.

**Estimating $\mathbb{E}_D$** Since $D = \mathcal{D}$ represents different draws of our training set, we estimate this expectation value as follows: we randomly subsample (without replacement) 90% of the training set. We perform this repeatedly over different seeds to form $t$ such trial sets $\mathcal{D}_k$, $k = 1, ..., t$ (see Figure 4). In the case of the small data limit, we subsample a smaller percentage of the training set (details are provided in later appendices). On each trial set, we train a neural network $h_{(L+1)}(\mathcal{D}_k)$. Importantly, we use the same seed to initialize the neural network weights on each trial set (per experiment, we fixed the initialization seed for two reasons, to minimize additional sources of randomness (in addition to the randomness introduced by data sampling – see for example works such as Adlam & Pennington (2020) and D'Ascoli et al. (2020) which study bias-variance decompositions with randomness coming from weight initialization) and to ensure that sub-predictors can be matched across trials to form the centroid (see for example Appendix C.7)). Thus, for a network with fixed hyperparameters, we train $t$ such networks $h_{(L+1)}(\mathcal{D}_k)$, $k = 1, ..., t$. We estimate the expectation $\mathbb{E}_D$ using an average.

**Results tensor** Additionally, we also have $d_L$ subpredictors in the neural network to consider: $q_{(i)}$, $i \in [d_L]$. Thus, combining everything, we construct a results tensor of shape $(t, d_L, s)$ with an entry in the tensor given by $q_{(i)}(x^{(j)}; \mathcal{D}_k)$; in other words, the output of subpredictor $i \in [d_L]$, trained on trial set $k \in [t]$, on test sample $j \in [s]$. The results tensor is then used to compute the relevant averages. To extract the output of a subpredictor on a test sample, we used Pytorch's forward hooks method.

The above procedure is repeated for each set of model hyperparameters. For example, for our shallow neural networks, we considered width-wise experiments for 7 widths. For each width, we used $t = 50$ trial sets and, therefore, trained 50 models. All in all, this results in 350 models that need to be trained to estimate a single bias-variance-diversity decomposition. Due to this fact and the hardware that we used (see Appendix B.2), we did not consider additional experimental details such as different choices of weight initialization or different train-test splits.

Finally, when performing the estimates, we found it necessary to use `torch.double()` precision (`torch.float64`) on the model weights, the output of the Pytorch forward hook, and the output of the neural network to avoid numerical issues.

### B.1.1 ESTIMATORS - LEAST SQUARES

In this section, we present the estimators of the bias, variance, and diversity terms for the case of square loss $\ell(x, y) = (x - y)^2$. For convenience, we restate the decomposition from Equation 10 here,

$$\mathbb{E}_D \mathbb{E}_{XY}[\ell(y, \bar{q})] =$$

$$\mathbb{E}_X \Bigg[ \underbrace{\mathbb{E}_{Y|X}[\ell(y, y^*)]}_{\text{noise}} + \underbrace{\frac{1}{d_L} \sum_{i=1}^{d_L} \ell(y^*, \mathring{q}_{(i)})}_{\text{average bias}} + \underbrace{\frac{1}{d_L} \sum_{i=1}^{d_L} \mathbb{E}_D[\ell(\mathring{q}_{(i)}, q_{(i)})]}_{\text{average variance}} - \underbrace{\frac{1}{d_L} \sum_{i=1}^{d_L} \mathbb{E}_D[\ell(\bar{q}, q_{(i)})]}_{\text{diversity}} \Bigg].$$

$$(21)$$

We estimate each term as follows:

$$\text{centroid:} \qquad \mathring{q}_{(i)}(x) = \mathbb{E}_D[q_{(i)}(x)] \approx \frac{1}{t} \sum_{k=1}^{t} q_{(i)}(x; \mathcal{D}_k) = \mathring{q}_{(i)}^{est}(x)$$

$$\text{bias:} \qquad \mathbb{E}_X \Bigg[ \frac{1}{d_L} \sum_{i=1}^{d_L} \ell(y^*, \mathring{q}_{(i)}) \Bigg] \approx \frac{1}{d_L} \frac{1}{s} \sum_{i=1}^{d_L} \sum_{j=1}^{s} \ell\big(y^{(j)}, \mathring{q}_{(i)}^{est}(x^{(j)})\big)$$

$$\text{variance:} \qquad \mathbb{E}_X \Bigg[ \frac{1}{d_L} \sum_{i=1}^{d_L} \mathbb{E}_D\big[\ell(\mathring{q}_{(i)}, q_{(i)})\big] \Bigg] \approx \frac{1}{t} \frac{1}{d_L} \frac{1}{s} \sum_{k=1}^{t} \sum_{i=1}^{d_L} \sum_{j=1}^{s} \ell\big(\mathring{q}_{(i)}^{est}(x^{(j)}), q_{(i)}(x^{(j)}; \mathcal{D}_k)\big)$$

$$\text{diversity:} \qquad \mathbb{E}_X \Bigg[ \frac{1}{d_L} \sum_{i=1}^{d_L} \mathbb{E}_D\big[\ell(\bar{q}, q_{(i)})\big] \Bigg] \approx \frac{1}{t} \frac{1}{d_L} \frac{1}{s} \sum_{k=1}^{t} \sum_{i=1}^{d_L} \sum_{j=1}^{s} \ell\big(\bar{q}(x^{(j)}; \mathcal{D}_k), q_{(i)}(x^{(j)}; \mathcal{D}_k)\big)$$

$$(22)$$

Note that, similar to (Kohavi & Wolpert, 1996; Domingos, 2000; Neal et al., 2019), we assume noiseless labels (in the sense that, for a given $x$, there is not a distribution of $y$ values) so that the noise term vanishes; in particular, we assume that $y^*$ can be approximated by the label $y$. This is a common simplification made when estimating bias-variance decompositions.

### B.1.2 ESTIMATORS - KL-DIVERGENCE/CROSS-ENTROPY

We next present the estimators for the KL-divergence/cross-entropy loss. Recall that we consider one-hot encoded target labels $y \in \mathbb{R}^C$. In this case, the KL-divergence reduces to the cross-entropy loss $K(y||\bar{q}) = -y \cdot \ln \bar{q} = \text{cross-entropy}(y, \bar{q})$ (see the discussion around Equation 12). As in the previous section, we will assume noiseless labels (not to be confused with the label corruption used in our experiments); in particular, the probability vector $y^*$ will be estimated by $y$. In this approximation, the decomposition, given by Equation 12, reduces to

$$\mathbb{E}_D \mathbb{E}_{XY}[K(y||\bar{q})] \approx$$

$$\mathbb{E}_X \Bigg[ \underbrace{-\frac{1}{d_L} \sum_{i=1}^{d_L} y \cdot \ln \mathring{q}_{(i)}}_{\text{average bias}} + \underbrace{\frac{1}{d_L} \sum_{i=1}^{d_L} \mathbb{E}_D[K(\mathring{q}_{(i)}||q_{(i)})]}_{\text{average variance}} - \underbrace{\frac{1}{d_L} \sum_{i=1}^{d_L} \mathbb{E}_D[K(\bar{q}||q_{(i)})]}_{\text{diversity}} \Bigg]. \qquad (23)$$

We estimate each term as follows:

Normalization: $\quad \mathring{Z}_i(x) = \sum_{c=1}^{C} \exp(\mathbb{E}_D[\ln q_{(i)}^c]) \approx \sum_{c=1}^{C} \exp(\frac{1}{t}\sum_{k=1}^{t} \ln q_{(i)}^c(x;\mathcal{D}_k)) = \mathring{Z}_i^{est}(x)$

Centroid: $\quad \mathring{q}_{(i)}(x) = \frac{1}{\mathring{Z}_i(x)} \exp(\mathbb{E}_D[\ln q_{(i)}]) \approx \frac{1}{\mathring{Z}_i^{est}(x)} \exp(\frac{1}{t}\sum_{k=1}^{t} \ln q_{(i)}(x;\mathcal{D}_k)) = \mathring{q}_{(i)}^{est}(x)$

Bias: $\quad \mathbb{E}_X\left[ -\frac{1}{d_L}\sum_{i=1}^{d_L} y \cdot \ln \mathring{q}_{(i)} \right] \approx -\frac{1}{d_L}\frac{1}{s}\sum_{j=1}^{s}\sum_{i=1}^{d_L} y^{(j)} \cdot \ln \mathring{q}_{(i)}^{est}(x^{(j)})$

Variance: $\quad \mathbb{E}_X\left[ \frac{1}{d_L}\sum_{i=1}^{d_L} \mathbb{E}_D\left[ K(\mathring{q}_{(i)}||q_{(i)}) \right] \right] \approx \frac{1}{t}\frac{1}{s}\frac{1}{d_L}\sum_{j=1}^{s}\sum_{i=1}^{d_L}\sum_{k=1}^{t} K\left(\mathring{q}_{(i)}^{est}(x^{(j)})||q_{(i)}(x^{(j)};\mathcal{D}_k)\right)$

Diversity: $\quad \mathbb{E}_X\left[ \frac{1}{d_L}\sum_{i=1}^{d_L} \mathbb{E}_D\left[ K(\bar{q}||q_{(i)}) \right] \right] \approx \frac{1}{t}\frac{1}{s}\frac{1}{d_L}\sum_{j=1}^{s}\sum_{i=1}^{d_L}\sum_{k=1}^{t} K\left(\bar{q}(x^{(j)};\mathcal{D}_k)||q_{(i)}(x^{(j)};\mathcal{D}_k)\right)$

$$(24)$$

### B.2 DETAILS AND HYPERPARAMETERS

In this section, we elaborate on the experimental setup used in Section 5. Furthermore, we report the hyperparameters used for the models in Appendix C.

**Double descent**  In order to elicit a clear double descent for the both the MNIST and CIFAR10 models, we introduce label noise by assigning a randomly selected different class label to a small percentage of the train set samples. All models are trained as close to zero train loss as possible over a fixed number of epochs, and no early stopping is performed. This is a similar setup to those used in prior work to elicit a clear model-wise double descent (Nakkiran et al., 2020; Somepalli et al., 2022).

**MLP architecture**  Our MNIST MLP consist of a single fully connected hidden layer followed by an output layer. We select the number of subpredictors (i.e., the width) in the range of $5$ to $5 \times 10^3$. This allows us to explore both the under-parameterized, critically-parameterized, and over-parameterized regime. This set uses the mean-field parameterization.

**CNN architecture**  Our CNN architecture has two ReLU-activated convolutional layers (using $3 \times 3$ kernels), with widths of $16$ and $k$, respectively. The second layer is followed by a global average pooling layer, which is finally followed by a fully connected output layer. Thus, to vary the number of subpredictors, we vary $k \in \{8, 16, 32, 64, 128, 512, 1024, 2048, 4096\}$. As mentioned, these models use the standard parameterization.

**CNN Subpredictors**  For standard parameterization, we construct our subpredictors by working backwards from the combiner. However, note that instead of each subpredictor consisting of fully connected layers as for MNIST, each subpredictor is now represented by a sequence of filters, a nonlinear activation, and a global averaging layer, which is then multiplied by an outgoing weight vector.

**Hyperparameters**  The exact hyperparameters for each model set is shown in Table 3, as well as those for the models used in Appendix C. Note that the top two rows ("CIFAR10 corrupt low data" and "MNIST full batch") correspond to those used in Section 5. Here follows additional details of the different settings:

- **Optimizer and optimization procedure**. All models are trained with (stochastic) gradient descent including momentum set to $0.9$. We do not employ early stopping in the traditional sense using a held-out set, although as the final model we select the epoch which reached the lowest *train loss*. This is simply to account for slight instabilities that can occur in the final epochs of training (e.g. a model suddenly 'forgetting' a sample).

- **Explicit regularization.** For all our experiments, we do not use any explicit regularization such as weight decay or Dropout. This is intentional as we are investigating sources of implicit regularization.

- **Learning rate warmup**. The learning rate for each set is as indicated in the table, although for some we first (linearly) warmup to this learning rate over a set number of epochs. This is indicated by the 'Warmup Epochs' column. It should also be noted that for the mean-field parameterization, the true learning rate for each model is given by the one specified times the number of subpredictors.

- **Learning rate scheduler**. Besides warmup, we also linearly decay the learning rate by multiplying it by $0.99$ every few epochs. The number of epochs used is indicated by the 'Scheduler Steps' column. The decay naturally only starts once warmup is complete. 'None' indicates that no scheduler is used.

- **Subsampling and label corruption**. As explained earlier in Section 5, we train 50 different models per width, where each is trained on a fraction of the train set. This fraction is indicated by the 'Subsample Size' column. Similarly, the 'Label Corruption' column indicates the fraction of the train set samples that are randomly assigned a different class label. Note that the label corruption is first applied to the entire train set before subsampling takes place.

**Computational resources**   Due to the large number of trials and experiments, the model sets in Table 3 total approximately $4\,500$ different MLPs and CNNs. To facilitate training such a large number of models, we trained several in parallel on either one or two Tesla V100 GPUs. Each set (around 350 models, depending on the set) required approximately 8 to 12 hours of training time.

Table 3: Hyperparameter settings for the different model sets considered.

| Group | Description | Param | Subsample Size | Label Corruption | Scheduler Steps | Learning Rate | Batch Size | Warmup Epochs | Max Epochs |
|---|---|---|---|---|---|---|---|---|---|
| CIFAR10 CNN | corrupt low data | Standard | 0.1 | 0.1 | 5 | 0.1 | 256 | 100 | 1000 |
| MNIST one layer | full batch | MFP | 0.9 | 0.2 | 50 | 1.2 | 54001 | 1000 | 10000 |
| | corrupt large data | MFP | 0.9 | 0.2 | 5 | 0.6 | 4096 | 100 | 1000 |
| | corrupt low data | MFP | 0.1 | 0.2 | 5 | 0.15 | 512 | 100 | 1000 |
| | clean large data | MFP | 0.9 | 0.0 | 5 | 0.6 | 4096 | 100 | 200 |
| | clean low data | MFP | 0.1 | 0.0 | 5 | 0.15 | 512 | 100 | 200 |
| | standard param | Standard | 0.9 | 0.0 | 5 | 0.4 | 4096 | 100 | 200 |
| MNIST two layer | fixed subpreds | MUP | 0.9 | 0.0 | None | 0.05 | 4096 | 100 | 500 |
| | varying subpreds | MUP | 0.9 | 0.0 | None | 0.05 | 4096 | 100 | 500 |
| Fashion MNIST | corrupt large data | MFP | 0.9 | 0.2 | 5 | 0.4 | 4096 | 100 | 1000 |
| California Housing | clean large data | MFP | 0.9 | 0.0 | 5 | 0.15 | 4096 | 0 | 2000 |
| | clean low data | MFP | 0.01 | 0.0 | None | 0.04 | 4096 | 0 | 1500 |
| | standard param | Standard | 0.9 | 0.0 | 5 | 0.0007 | 2048 | 0 | 1000 |

## C  ADDITIONAL EXPERIMENTAL RESULTS

In this section, we provide further experimental evidence that diversity acts as an implicit regularizer. Additionally, we provide experimental results for other choices of parameterization.

The subsections of this appendix are organized as follows. We first consider three additional experimental configurations of MLPs trained on MNIST: In Appendix C.1, we study the effects of full batch training versus mini-batch training, before analyzing the small versus large data limit in Appendix C.2. This is then followed by Appendix C.3, where we consider two-hidden layer neural network using $\mu P$ parameterization. After this, we move on to additional datasets and consider Fashion MNIST in Appendix C.4, and the regression task consisting of the California housing dataset in Appendix C.5. We then consider results for both MNIST and the California housing dataset when using the standard parameterization in Appendix C.6. Finally, in Appendix C.7, we analyze the problem of subpredictor allocation in a neural network.

## C.1 THE EFFECT OF MINI-BATCH SGD

As discussed in Section 1, there are other sources of implicit regularization due to the learning process. For example, it has been shown that SGD implicitly regularizes the model through a modified loss function of the infinitesimal version; in particular, the loss function receives two modifications, namely, a term that penalizes the norm of the full batch gradient and a term that penalizes the variability of the mini-batch gradients (Smith et al., 2021). In Section 5, we reduced the effects of this additional implicit regularization by using full batch training. This allowed us to study the implicit regularization of diversity in the absence of the additional source of implicit regularization [5]. In this section, we extend this result to mini-batch training, which allows us to verify whether the same trends hold when the implicit regularization of SGD is also present. It also provides additional insight into the effects of using mini-batch versus full batch training on the bias, variance, and diversity of the neural networks.

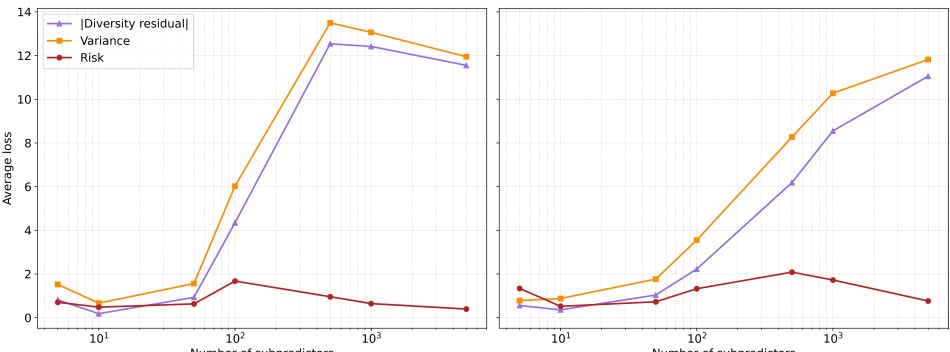

Figure 5: The variance, risk, and absolute value of the diversity residual for label corrupt large data MNIST using mini-batch training (left) versus full batch training (right). Note that, for 5 subpredictors in the right plot, the risk is higher than the variance and the diversity. This is due to the fact that the diversity residual starts off as positive and then, for more subpredictors, becomes negative.

**Setup**    We train a one-layer MLP on MNIST with mini-batch SGD using the same setup as that of the full-batch MNIST MLP presented in Section 5. We then estimate the bias, variance, and diversity and compare the two sets of results. See the "MNIST one layer - corrupt large data" row in Table 3 for hyperparameter details.

**Results**    Figure 5 shows the variance, risk, and absolute value of the diversity residual for mini-batch training (left) and full batch training (right) on label corrupted MNIST. For the case of mini-batch training, at the critical regime (approximately 100 subpredictors), the neural network interpolates the training data (zero training loss) and, similarly, for the models in the overparameterized regime. In contrast, for full batch training, the models struggle to interpolate the training data, with the interpolation point being shifted to 500 subpredictors. This is mainly due to the optimization being unstable for full batch training (Geiping et al., 2022).

For both the mini-batch training and full batch training, we observe that the variance rapidly increases once a model interpolates the training data, as previously reported. However, for mini-batch training, we also observe that both the variance and diversity starts to naturally decrease after 500 subpredictors where, for full batch training, both keep increasing. The decrease of the variance error for mini-batch training is likely due to the additional implicit regularization. However, despite this decrease, we also observe that the gap between the variance and the diversity residual decreases when we use mini-batch training (and thus the risk is lower). This suggests that mini-batch training has an overall positive effect on the diversity of the subpredictors. We believe that an interesting direction of research would be to better understand the connection between the diversity of the subpredictors and mini-batch training.

---

[5]However, note there are still other sources of implicit regularization present even when using full batch training – see for example Barrett & Dherin (2021).

## C.2 VARYING DATA SIZE AND LABEL CORRUPTION

In Section 5, we studied the effect of diversity on neural networks trained on label corrupted MNIST. The label corruption allows for a visible second descent of the risk curve. This allowed us to study the effect of diversity when transitioning between the underparameterized regime, the critical regime, and the overparameterized regime. In this section, we extend these results and consider additional configurations, namely 1) the low data limit, and 2) MNIST without label corruption (clean MNIST). By comparing these variants we are able to determine the interplay between the estimated bias, variance, and diversity in different training scenarios. Importantly, it allows us to verify whether the diversity still implicitly regularizes the variance under different training configurations.

**Setup** We compare the following four sets of models:

- **Label corrupted with large data**. The same (mini-batch SGD) set from Appendix C.1, where each trial model is trained on a random 90% subset of the MNSIT training set, and 20% of the samples have a randomly assigned different label.

- **Label corrupted with small data**. The same as above, but now each trial is trained on a mere 10% subset of the training data. We expect this modification to significantly increase the variance and increase the risk.

- **Clean with large data**. For this set, we do not introduce any label noise, and each model is trained on a 90% subset of the training data. With clean data, we no longer expect to observe a double descent on the risk curve. This allows us to determine the effects of the diversity regularization when the risk is a monotonically decreasing function of the number of subpredictors. Additionally, it allows us to determine whether the increased diversity observed in Section 5 is merely a side-effect of noisy labels.

- **Clean with small data**. The clean counterpart to the label corrupted small dataset of models.

For all of these sets, the training setup is kept the same as described earlier in Section 5, except that we employ mini-batch SGD and adjust the hyperparameters to ensure proper convergence. See Table 3 for the exact settings. Similarly, we again estimate the bias, variance, and diversity in the same fashion. The results of this analysis for all four sets is shown in Figure 6 with a shared y-axis.

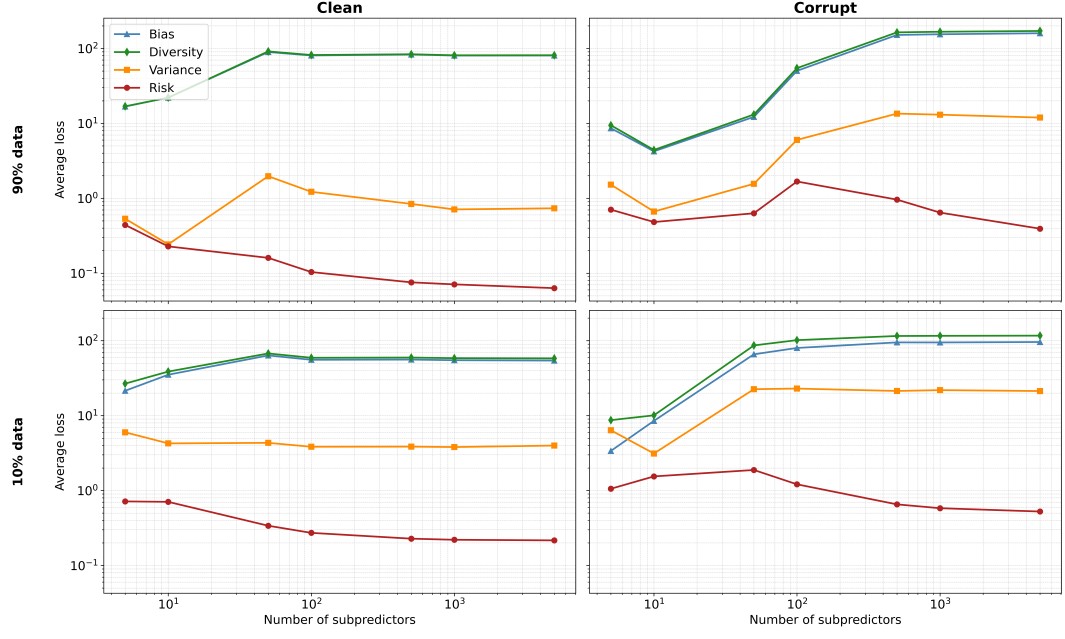

Figure 6: The estimated bias-variance-diversity decomposition for clean (left) and label corrupted (right) MNIST with large (top) and small (bottom) data. Note that the y-axis is shared across all plots.

**Results**    We first compare the risk curves of these four sets. When comparing the clean models with the corrupt (left to right), it is clear that the label corruption results in increased risk, and also the familiar curve which shows a double descent. As expected, the clean models do not show this double descent and the risk decreases as the number of subpredictors is increased (it is well established in prior work that a double descent is typically not easily observed without introducing label corruption (Nakkiran et al., 2020; Somepalli et al., 2022)). Comparing the large data models versus the small data models, it is evident that the risk is increased when the size of the dataset is reduced. For the label corrupted models, we also observe that the point of highest risk has shifted from 100 subpredictors with the large data to 50 subpredictors with the small data. This too is to be expected, as less capacity is required to fit a smaller number of samples, i.e. the interpolation regime has shifted.

When considering the diversity curves, we observe that all four sets shows similar behavior to that observed earlier: The diversity closely tracks the bias error, and the diversity is larger than this bias. In order to do a more fine-grained comparison, we again plot the absolute value of the diversity residual and the variance instead of all three terms separately. This is shown in Figure 7. Note that the y-axis is now on a linear scale.

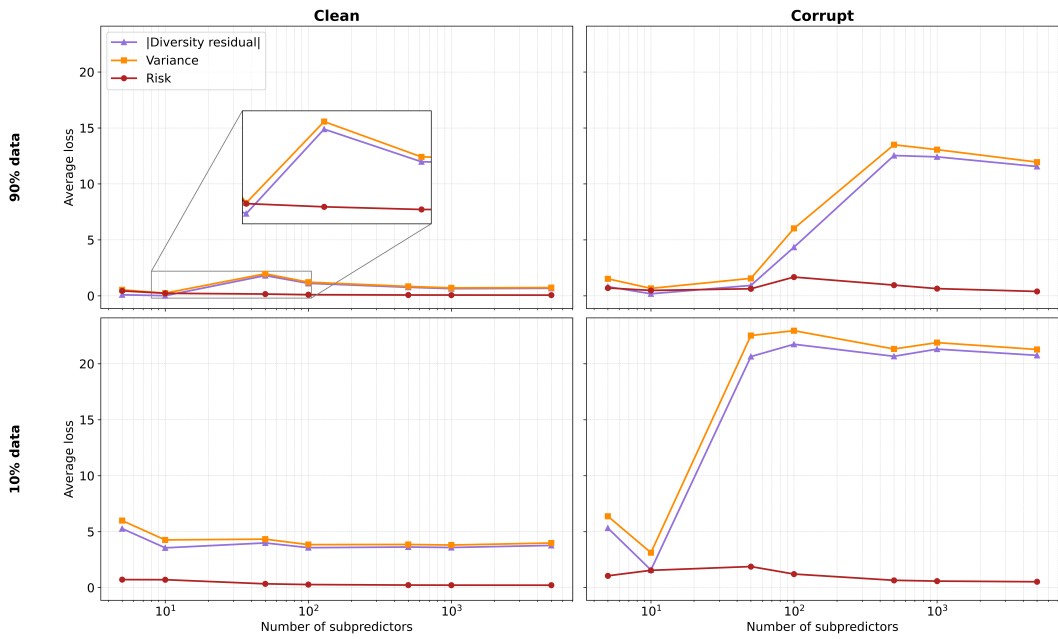

Figure 7: The estimated diversity residual decomposition for clean (left) and label corrupted (right) MNIST with large (top) and small (bottom) data. Note that the y-axis is shared across all plots and on a linear scale.

Armed with this visualization, let us once again consider the differences in risk observed between the different sets:

- **Clean versus corrupt** (left to right comparison): For the corrupt sets, we observe that the variance is significantly larger than their clean counterparts. Interestingly, we observe that the diversity residual is also significantly larger. However, the difference in risk between the two sets can be explained by the fact that the diversity for the clean models appears to grow faster in comparison to the variance, and therefore the 'gap' between the two terms is smaller.

- **Large data versus small data** (top to bottom comparison): Similarly to above, we observe increased variance for the small data models. Again, we also observe a corresponding increase in the diversity residual, however the gap between the two terms remains larger for the small data models.

It is also interesting to note that for the clean large dataset of models, we still observe a rapid increase in the diversity and variance as the model width transitions from the under-parameterized to the over-parameterized regime. This is highlighted by the zoomed region in the top left of Figure 7. This shows that the regularization effect of the diversity is not merely an artifact of noisy labels. Taken together, we observe the following for all four sets: 1) the diversity closely tracks the bias, 2) the diversity grows as the variance grows (i.e. the diversity regularizes the variance), and 3) the relative size between the variance and diversity explains the observed risk. In conclusion, these results support our findings expressed in Section 5.

## C.3 Two-hidden layer MLPs

In terms of MLPs, we have relied on shallow networks to demonstrate the bias-variance-diversity decompositions and the effect of diversity on the variance error. In this section, we study the influence of depth in the neural network. For deep neural networks, we used the $\mu P$ parameterization scheme. We consider a two-hidden layer neural network with hidden widths $d_1, d_2 \in \mathbb{N}$. For this parameterization, we observe that the subpredictors contains a factor of $\sqrt{d_2}$ in their definitions. Hence, the subpredictor outputs might diverge as a function of width in the last hidden layer.

**Shallow versus Deep**  Recall that the subpredictors are determined by the hidden nodes of the last hidden layer, multiplied by an outgoing weight. In contrast to the shallow neural network, the subpredictors now share weights (the weights that extend from the input layer to the first hidden layer). Thus, the diversity of the subpredictors may be impacted. Additionally, for a fixed second hidden layer width $d_2$, we can now vary the capacity of each subpredictor by varying the width $d_1$ of the first hidden layer. This may impact the bias and variance error of each subpredictor.

**Setup**  We consider two experiments for a two-hidden layer neural network on clean MNIST (hyperparameter details can be found in Appendix B.2). In the first experiment, we fix the number of subpredictors in the neural network by fixing $d_2 = 100$ and we vary the width of the first hidden layer $d_1$ over values 5, 10, 50, 100, 500, 1000. For the second experiment, we vary the number of subpredictors through $d_2$ while keeping the width of the first layer fixed to $d_1 = 100$, which can be considered as the rough MLP equivalent of our CNN experiment in Section 5.

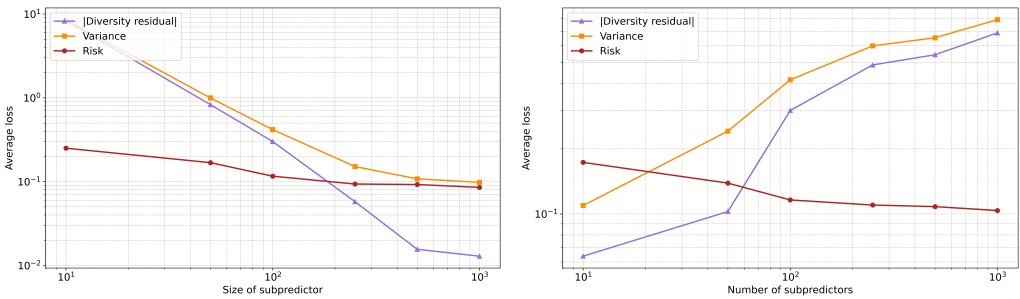

Figure 8: The variance, risk, and absolute value of the diversity residual for two layer MLPs trained on MNIST. Left: Fixed subpredictors - First layer with a varying capacity and second layer with a fixed 100 subpredictors. Right: Fixed capacity - First layer with a fixed width of 100 and a second layer with a varying number of subpredictors.

**Results**  Figure 8 shows the variance, risk, and absolute value of the diversity residual for the networks with a fixed number of subpredictors (left) and fixed capacity (right). We note that the fixed capacity two layer networks show a curve similar to what we've observed earlier: the variance increases as the number of subpredictors increases, but this is matched by a larger increase in the diversity residual and the risk decreases. On the other hand, we see a different trend on the left side of Figure 8. We observe that, for the fixed number of subpredictor networks, that both the variance and diversity decreases as the capacity of the subpredictors increases. We suspect that this is likely due to the fact that the subpredictors now share weights among themselves which causes the subpredictors to be more correlated with each other and, hence, less diverse. Despite this, we note that the diversity

residual remains positive and therefore still regularizes the variance for subpredictors larger than 100 nodes (note that the log scale of the graph exaggerates the difference between the variance error and the diversity, especially for smaller values).

Although we have captured the effect of diversity for subpredictors defined through the final layer of the neural network, we hypothesize that there are further hidden diversity effects when the network has depth, which may help explain why the variance naturally decreases when we vary the width of the first hidden layer. We believe that the identifications we used in Section 4.2 can, in principle, be extended to the level of subpredictors by treating a subpredictor as a combiner and repeating the procedure to identify new subpredictors that are defined by the first hidden layer.

### C.4    FASHION MNIST

In this section, we consider one-layer MLPs trained on the fashion MNIST dataset (FMNIST) (Xiao et al., 2017) This allows us to verify our MNIST results on a different dataset.

**Setup**    We again train single hidden layer neural networks using the mean-field parameterization and vary the number of subpredictors by varying the width of the neural network. We use 20% label corruption to induce a double descent. Additional hyperparameter details concerning the training of the models can be found in Table 3.

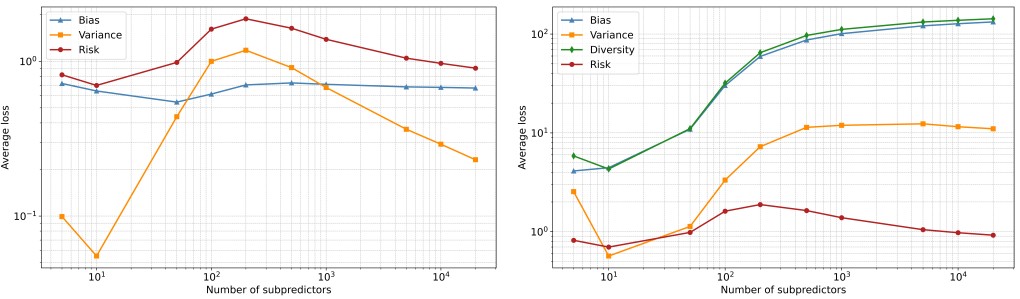

Figure 9: The estimated bias-variance decomposition (left) and bias-variance-diversity decomposition (right) for models with an increasing number of subpredictors trained on label corrupted Fashion MNIST. The estimates are calculated over 50 trials for each model, where each trial is trained on a randomly sampled 90%.

**Results**    Figure 9 shows the usual bias-variance decomposition (left) and the bias-variance-diversity decomposition (right). Similar to the MNIST experiments, for the usual bias-variance decomposition, we observe that initially the bias decreases and the variance increases to produce a U-shaped risk curve. Then, at the critical interpolation point at 500 subpredictors, the risk curve produces a peak and starts to display a second descent. Similarly, we again find that the variance inexplicably decreases after the interpolation point. Interestingly, we also find that the bias decreases up to 50 subpredictors, then increases up to 500 subpredictors, and then decreases slowly again. In contrast, for the bias-variance-diversity decomposition, we observe the same pattern as with our other experiments, namely, that bias, variance, and diversity tends to increase as we vary the number of subpredictors.

In Figure 10, we plot the absolute value of the diversity residual. Similar to our other experiments, we observe that the diversity residual tracks the variance error and regularizes it to produce the resulting risk curve.

### C.5    CALIFORNIA HOUSING DATASET

In this appendix, we confirm our bias-variance-diversity decomposition for a regression task using squared error loss as well as the implicit regularization due to diversity. In particular, we use the popular California Housing dataset (Pace & Barry, 1997). We performed two types of experiments where we used the full training set as well as a small data limit, similar to Appendix C.2, although we

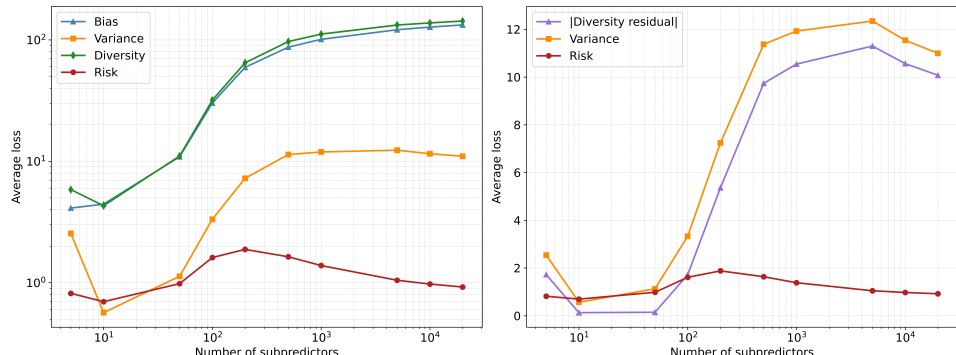

Figure 10: The estimated bias-variance-diversity decomposition (left) and absolute value of the diversity residual (right) for models with an increasing number of subpredictors trained on label corrupted Fashion MNIST. The estimates are calculated over 50 trials for each model, where each trial is trained on a randomly sampled 90%.

do not consider artificial label corruption. In the small data limit, most of our models are significantly in the overparameterized regime.

### C.5.1 DATA PREPARATION AND MODELS

**Data**  We used a train, validation, and test split of 0.6, 0.1, and 0.3, respectively. In particular, we aimed to keep the test set large in order to ensure good estimates of the bias-variance-diversity decompositions. Categorical features were one-hot encoded. Using the mean and standard deviation from the training set, we z-normalized all numerical features on both the train, validation, and test set. Any missing values were replaced with the mean.

For the trial sets used in the estimates (see Appendix B.1), we randomly subsampled 90% of the training set over different trials. Similarly, for our experiments in the small data limit, we randomly subsampled 1% of the training set.

**Models**  In both the large data and small data experiments, we trained single hidden layer neural networks of widths 5, 10, 50, 100, 500, 1000, and 5000, using the mean-field parameterization. Importantly, for a fixed width, we used the same seed to initialize the model weights over each trial set. The training details for the experiments are given in Appendix B.2. Similar to the double descent experiments on MNIST, we aimed to minimize the training loss as far as possible.

### C.5.2 DECOMPOSITIONS

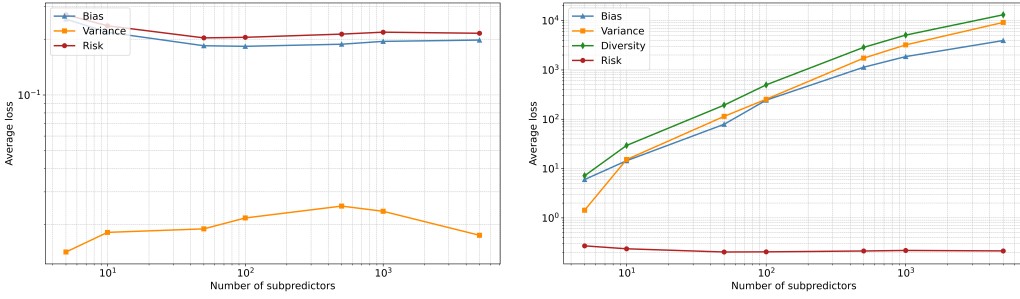

Figure 11: The estimated bias-variance decomposition (left) and bias-variance-diversity decomposition (right) for models with an increasing number of subpredictors trained on California Housing. The estimates are calculated over 50 trials for each model, where each trial is trained on a randomly sampled 90%.

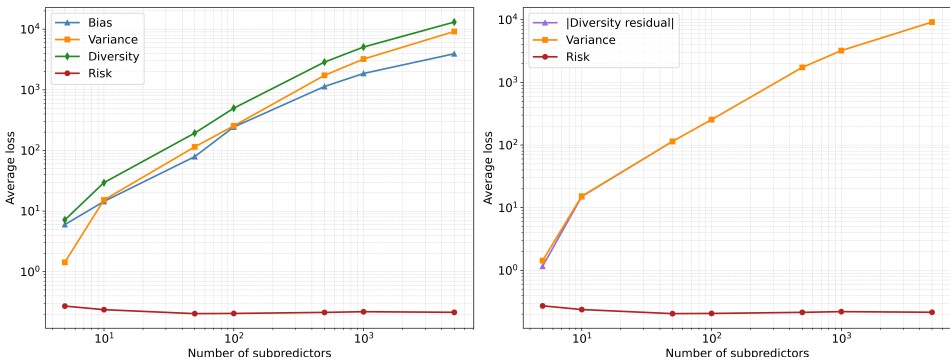

Figure 12: The estimated bias-variance-diversity decomposition (left) and absolute value of the diversity residual (right) for models with an increasing number of subpredictors trained on California Housing. The estimates are calculated over 50 trials for each model, where each trial is trained on a randomly sampled 90%.

Similar to our MNIST results, we observe the same patterns in the decompositions for the California Housing dataset. Figure 11 shows the usual bias-variance decomposition versus the new bias-variance-diversity decomposition for the large dataset of models. Interestingly, although difficult to see in the plot, we observe a U-shaped curve for the risk followed by a second descent (a clear plot of the risk is shown in the left image of Figure 14) with the critical point at approximately 1000 subpredictors. This is likely due to inherent noise in the dataset. As in the MNIST double descent experiments, for the usual decomposition (left figure) we again see a decreasing bias, with the variance increasing, until about 500 subpredictors where it starts to decrease again. In contrast, for the new decomposition (right figure), we see that the bias and variance errors of the subpredictors increase as we increase width but start to plateau at 5000 subpredictors. At the same time, the diversity term also increases and its magnitude is larger than the bias and variance errors. Similar to our other experiments, to see the effect of diversity on the variance error, we plot the absolute value of the residual diversity in Figure 12. Although difficult to see in the plot, the residual diversity strongly regularizes the variance error, which results in the final observed risk curve. A slightly more clear plot can be found in the left figure of Figure 14.

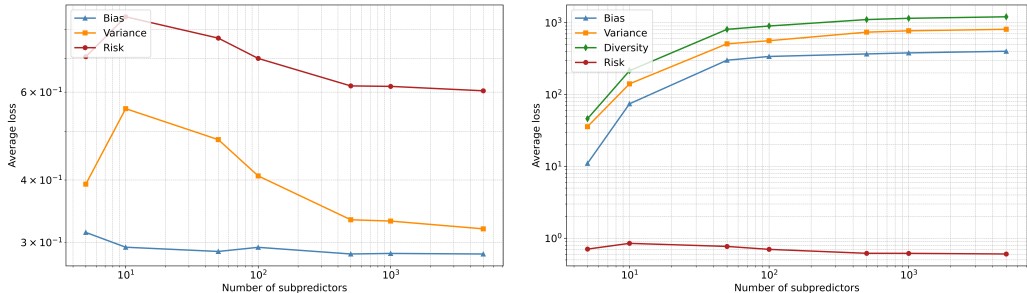

Figure 13: The estimated bias-variance decomposition (left) and bias-variance-diversity decomposition (right) for models with an increasing number of subpredictors trained on a small subset of California Housing. The estimates are calculated over 50 trials for each model, where each trial is trained on a randomly sampled 1%.

**Small data limit** The patterns remain similar for the small data limit. Figure 13 shows the usual bias-variance decomposition and the new bias-variance-diversity decomposition. As with the large data case, we again see a U-shaped risk curve, with an increase in the risk up to 10 subpredictors, followed by a decrease in the risk. As before, for the left figure (bias-variance) we see that the bias steadily decreases while the variance increases up to 10 subpredictors and then starts to decrease again. In contrast, for the right figure (bias-variance-diversity), we see that the bias and variance both

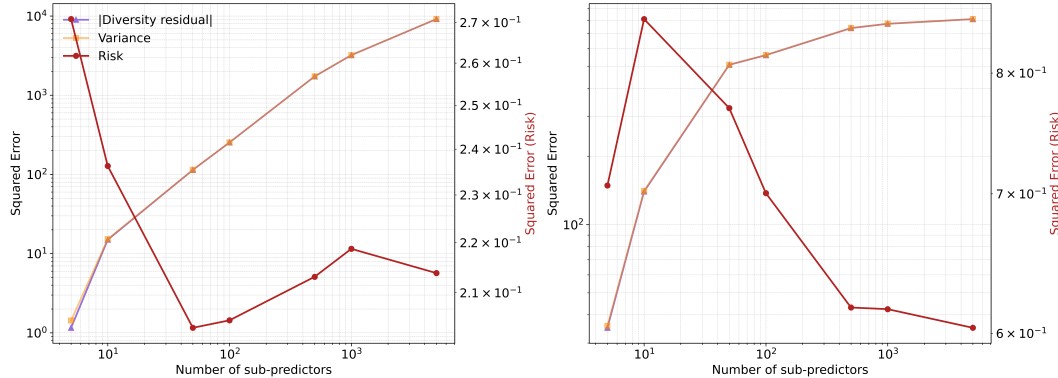

Figure 14: Diversity residual plot for the large data limit (left) and the small data limit (right) of the California Housing dataset. In both cases, we see that the diversity residual strongly regularizes the variance error. Note, due to the differences in magnitudes, we use a twin-axis plot with the left axis indicating the squared error for the residual and the variance, and the right axis indicating the error for the risk.

increase but eventually plateaus after approximately 50 subpredictors. The diversity closely tracks the bias and variance errors and, as before, is greater in magnitude.

In Figure 14, we compare the absolute value of the diversity residuals for the large data limit (left figure) and the small data limit (right figure) on a twin-axis plot (left axis is squared error for the variance and the residual, while the right axis is the squared error for the risk). We observe that the diversity residual strongly regularizes the variance in both cases.

## C.6 STANDARD PARAMETERIZATION

We have predominantly conducted experiments using the mean-field parameterization, while we consider $\mu P$ parameterization in Appendix C.3 as part of the network depth experiments. In this appendix, we repeat the same one-layer MLP experiments on clean MNIST and California Housing for the standard parameterization. Notably, the standard parameterization has an explicit factor of $d_1$ that appears in the definition of the subpredictor. In the large width limit, the subpredictors might, therefore, diverge.

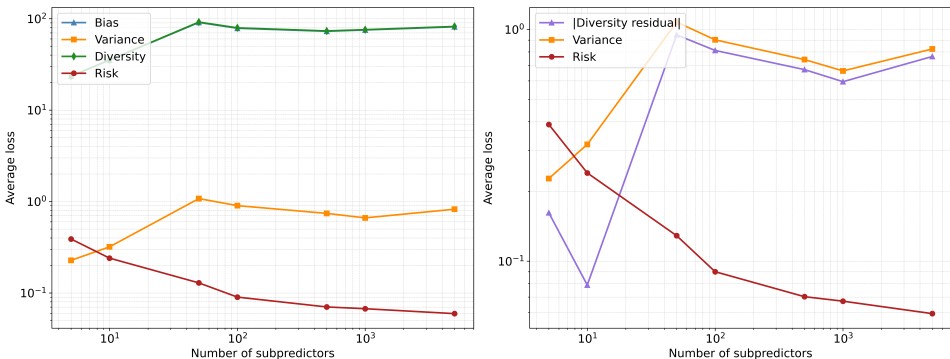

Figure 15: The estimated bias-variance-diversity decomposition (left) and the absolute value of the diversity residual (right) for models with an increasing number of subpredictors trained on clean MNIST using standard parameterization. The estimates are calculated over 50 trials for each model, where each trial is trained on a randomly sampled 90%. Note that the residual is initially positive at 5 subpredictors and then becomes negative for 10 subpredictors and higher.

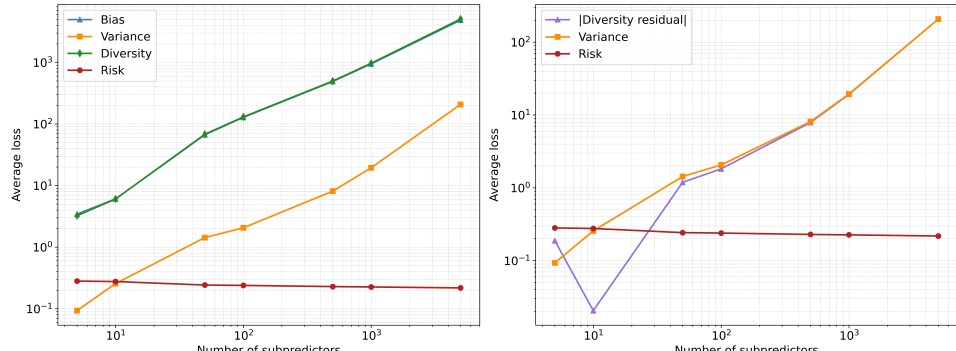

Figure 16: The estimated bias-variance-diversity decomposition (left) and the absolute value of the diversity residual (right) for models with an increasing number of subpredictors trained on California Housing using standard parameterization. The estimates are calculated over 50 trials for each model, where each trial is trained on a randomly sampled $90\%$. Note that the residual is initially positive at 5 subpredictors and then becomes negative for 10 subpredictors and higher.

**Setup**    We train the 'clean large data' variants for the MNIST and California Housing datasets of Appendices C.2 and C.5, respectively, using the standard parameterization. See the 'standard param' rows of Table 3 for details.

**Results - MNIST**    In Figure 15 we show the estimated bias-variance-diversity decomposition (left) and the absolute value of the diversity residual (right) for the standard parameterization MNIST models. We see that, despite the explicit factor of $d_1$ that appears in the subpredictors for the standard parameterization, the bias, variance, and diversity (left) does not diverge as width is increased. Instead, all three terms show an increase until 50 subpredictors and then plateaus for higher subpredictors. As in the MFP case, we observe that the diversity residual closely tracks and regularizes the variance error.

**Results - California Housing**    In Figure 16, we again plot the bias-variance-diversity decomposition (left) and the absolute value of the diversity residual (right). We now see that the bias, variance, and diversity terms increase as a function of the number of subpredictors without producing a plateau as in our other experiments. This suggests that the explicit $d_1$ factor that appears in the definition of the standard parameterization subpredictors starts to dominate as the width of the neural network (in other words, $d_1$) increases. However, despite the diverging variance, we still find that the diversity residual closely tracks and regularizes the variance error.

## C.7    SUBPREDICTOR ALLOCATION

For each model architecture, a centroid is estimated per subpredictor across trials. Given that nodes in a dense layer can occur in any order without affecting model output, how do we know the correct subpredictors are matched across trials? From Theorem 1, sub-predictor allocation will not affect the value of the diversity estimate, but could have an impact on the balance between the bias and variance estimates. We therefore explore additonal ways to allocate subpredictors to centroids (across trials) and measure the effect of these allocations on variance: the best allocation is expected to produce the lowest variance. Specifically, we consider different measures of similarity, related to either the individual predictions or the weight structure of subpredictors.

**Setup**    Trials are re-organised in a sequential manner. When re-organising the subpredictors in a single trial, a prototype trial is first constructed. The simiarlity metric is then applied to all $d_l$ subpredictors, across all samples, resulting in a $d_l \times d_l$ matrix. Using this matrix, the best-matching estimators are paired in a greedy manner, resulting in a new ordering applied to the specific trial.

We explore different setup variants using the 'small data' California Housing regression task models. Among others, we investigate:

- **Prototype:** Selecting either the mean over trials or the first trial as the prototype.

- **Prediction matching:** The predictions for all samples are compared using either correlation among predictions, or prediction loss.

- **Function distance:** The weight structure of each sub-predictor is used directly, to determine the function that the sub-predictor applies to the input. This function vector is compared using cosine similarity. Either the whole weight structure is considered (*full*), or only the first layer ($L1$).

**Results** We show the estimated variance using these allocations in Figure 17. Similar results were obtained using additional variants, such as using the magnitude (rather than signed value) and considering different forms or normalization: trends are similar but no variant obtains a better predictor than the initial ordering. At larger layers, this becomes very similar to the weight-based ordering if only the first layer is considered. We find that the best indicator is the value with which the sub-predictor was initialized. This is the allocation used in all reported experiments.

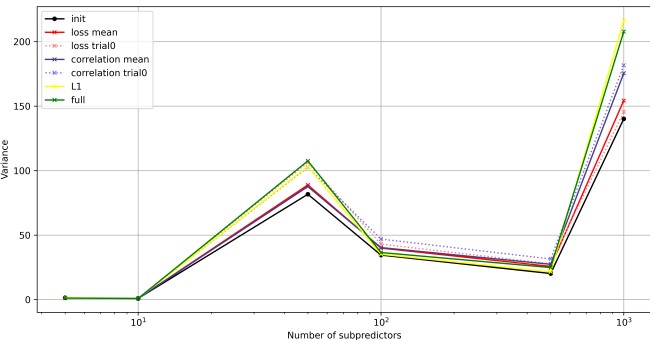

Figure 17: Estimated variance as subpredictors are re-ordered using different metrics: *init* is the original ordering; *correlation* or *loss* indicates different ways to measure similarity; *trial0* or *mean* indicates different choices of prototype when using predictions over samples, and *full* or *L1* indicate that the weight structure was used, either the full structure or only the first layer. Note that the variance of *init* (black) is always lowest.

