# OpenReview forum: "Implicit Regularization Through Hidden Diversity in Neural Networks"
_ICLR.cc/2026/Conference — Submitted to ICLR 2026_

### Official Review · Reviewer_ZfaN · 2025-10-28

**Soundness:** 4
**Presentation:** 1
**Contribution:** 3
**Rating:** 2
**Confidence:** 3

**Summary:**

- The paper proposes writing a single neural network with a dense final layer, as a weighted sum of subpredictors, where the weights are the rows of the logit layer scaled by a scalar multiplier depending on the initialization scheme.
- It applies Wood et al.'s decomposition to these subpredictors, viewing them as an ensemble of learners that exhibit a "diversity" term.
- The experiments suggest that, if I understood correctly, the risk of a single neural network can be estimated by computing the bias, variance, and diversity terms on the subpredictors.

**Strengths:**

If I understood correctly, this work presents a novel application of Wood et al.'s decomposition to a single neural network, offering interesting insights into the generalization properties of deep nets. This is particularly relevant given the widespread use of deep learning models and the lack of consensus on the mechanisms underlying good generalization. It provides an additional perspective on this important topic, using quantities that are simple to compute.

**Weaknesses:**

- The paper appears to be a shortened version of a journal article, which may be better suited for a journal submission. For an ICLR submission, more careful editorial choice could be applied to decide what content belongs in the main body versus the appendix.
- The main body of the paper is not fully self-contained, with missing details. For example, it is not clearly stated what is plotted in Figure 3, which I believe is the primary experimental result. As a result, I largely had to guess what exactly is the takeaway, what and how exactly are the quantities computed.
- There is an excessive focus on irrelevant details, such as in section 3.2, where the general case for centroids is explained, despite only requiring a simple discrete weighted average  for both MSE loss and KL divergence in the rest of the paper.
- The paper lacks explicit actionable takeaways. For instance, it could be beneficial to directly incorporate diversity as a regularizer during training, making this concept more practical and accessible.

**Questions:**

1. In Section 3.3, Theorem 1: What is y, as defined in Section 3.1? What are q_s ? Are they densities as described in Section 3.2?
2. Line 261: Why is it necessary for the coefficients to satisfy \sum \beta = 1 ? From Section 3 alone, it is not clear why the ( p(q_{(i)}) ) coefficients should form a valid probability mass function rather than any arbitrary set of weights. Is it actually relevant in the context of the paper?
3. Line 276: h_{(1)} depends on the training set, but it is never explicitly mentioned how up to that point. I assume the models are trained, but this should be clearly stated. Are different training sets used since, in the expectations further below (Eq. 10), D is a random variable?
4. What exactly is plotted in Figure 3 (right)? How are all the terms computed? Is it (option 1) the same training subsets as in Figure 3 (left), or (option 2) are these estimated quantities over subpredictors? If option 2, are the subpredictors trained on the entire training set, while the estimators in the left figures are trained only on subsets? Is there a scaling factor between quantities on the left and their estimators on the right ?

Minor comments:
===============
5. Line 256: q^i versus q_{(i)} — why the change from subscript to superscript?
6. Section 3.1: The indices i are used both for training examples and subpredictors, which causes confusion (e.g., the formula for R_{emp} on line 117 does not make sense). It is also used in d_i, where I assume it refers to input space dimension.
7. Line 376: "It is known that width is the primary factor for good network performance" — this should be toned down, as for example, large language models often require increased depth rather than just increased width.
8. Line 849 (Appendix): "Importantly, we use the same seed to initialize the neural network weights on each trial set." If this is an important point, it should be discussed in the main body: what happens when seeds are not fixed?
9. Experiments are a bit toyish. Ideally, it would be interesting to include experiments on more realistic datasets, even just a ResNet on CIFAR-10 rather than fully convolutional neural networks.

---

> ### Author Response · Authors · 2025-11-20
>
> We thank the reviewer for the comments and the questions. We appreciate that the reviewer finds that our paper provides an interesting perspective on the important topic of mechanisms underlying a neural network’s ability to generalize.
>
> # Weaknesses
>
> >The paper appears to be a shortened version of a journal article, which may be better suited for a journal submission. For an ICLR submission, more careful editorial choice could be applied to decide what content belongs in the main body versus the appendix.
>
> **Response (1)**: We developed this paper as a conference paper (there is no other version). Our main paper contains the main claims, results and the background necessary to follow our arguments; the appendix includes further experiments and expands on theoretical details for interested readers. Please advise if there are specific aspects that you would prefer to see moved to the main paper - the page limit increases from 9 to 10 pages during rebuttal, so we do have some space available.
>
> >The main body of the paper is not fully self-contained, with missing details. For example, it is not clearly stated what is plotted in Figure 3, which I believe is the primary experimental result. As a result, I largely had to guess what exactly is the takeaway, what and how exactly are the quantities computed.
>
> **Response (2)**: With respect to Figure 3, the plots show how the bias-variance-diversity decomposition of the expected risk (Equations (10) and (12)) compares to the classical decomposition of Geman et al. (1992) for a single neural network. This is explained in  Section 5: see the paragraphs leading up to and including the paragraph on “Implicit regularization”.
>
> The takeaway to Figure 3 is discussed in the paragraphs on “Implicit regularization” in Section 5. In terms of how the quantities are computed, we provide all details in Appendix B (kindly see the paragraph “Setup” in Section 5).
>
> We believe this should make it clear what is plotted in Figure 3.
>
> >There is an excessive focus on irrelevant details, such as in section 3.2, where the general case for centroids is explained, despite only requiring a simple discrete weighted average for both MSE loss and KL divergence in the rest of the paper.
>
> **Response (3)**: For the background theory introduced in Section 3, the centroid plays a critical role: for a given loss function, it computes both the correct form for the expected model (over training sets) necessary to perform the decompositions and, as a special case of the centroid, computes the correct form of the combiner necessary for Theorem 1 to work. Excluding the discussion on the centroid will undermine their important role in Theorem 1. Additionally, since readers may not be familiar with the concept of the centroid, we found it prudent to discuss it. Also note that the centroids computed in Lemma 1 and 2 are directly used in the construction of the implicit ensemble detailed in Section 4 and Appendix A.3.1. Finally, note that the decomposition is valid for a broad family of loss functions which may result in different combiners (centroids) than the two examples we used in the paper (square loss and KL divergence).
>
> >The paper lacks explicit actionable takeaways. For instance, it could be beneficial to directly incorporate diversity as a regularizer during training, making this concept more practical and accessible.
>
> **Response (4)**: Our paper, whose main goal is to propose and demonstrate a new source of implicit regularization in neural networks, follows in the same spirit as other theoretical works on implicit regularization in neural networks. See for example, Neyshabur et al. (2015), Gunesekar et al. (2017), Arora et al. (2019), and other references in our Section 2. Notably, many of these works do not explicitly provide actionable takeaways in the sense that is proposed in the comment. Instead, they propose a study of a mechanism behind implicit regularization and then experimentally validate their findings.
>
> Nonetheless, we agree that such actionable takeaways are important and relevant – for example, we conjecture that regularizers such as Dropout likely work as they are increasing the diversity between subpredictors. Since such investigations would likely make our main paper too long, we postpone it as interesting future avenues of research (we have added this to a ‘Future work paragraph – see line 492).

---

> > ### Author Response · Authors · 2025-11-20
> >
> > (continued)
> >
> > # Questions
> >
> > >In Section 3.3, Theorem 1: What is y, as defined in Section 3.1? What are q_s ? Are they densities as described in Section 3.2?
> >
> > **Response (5)**: If we understand the question correctly: since we are considering a supervised learning problem (see Section 3.1), $y$ is a target variable of the problem that is paired with a corresponding input $x$.
> >
> > The $q$’s are the members of the ensemble (with $\mathring{q}$ forming the centroid with respect to $D$ and $\bar{q}$ forming the combiner of the ensemble).
> >
> > Depending on the supervised learning problem, they may be densities (as in the case of Heskes (1998) for example) or continuous real variables (like in the case for a standard regression task such as the square loss example that we consider) for example.
> >
> > >Line 261: Why is it necessary for the coefficients to satisfy \sum \beta = 1 ? From Section 3 alone, it is not clear why the ( p(q_{(i)}) ) coefficients should form a valid probability mass function rather than any arbitrary set of weights. Is it actually relevant in the context of the paper?
> >
> > **Response (6)**: Consider Definition 1 in Section 3.2 for some loss function. In the definition, we have an expectation computed over the random variable $T$. Notably, the expectation is an integral of the loss function weighted by some density. This (normalized) density needs to integrate to unity. Now suppose $T$ was a discrete random variable. Then, we replace the expectation $\mathbb{E}$ with the weighted average $\sum_{i=1}^{m}p(t_i)$. Now, we require that the weights $p(t_i)$ sums to unity. This is precisely what we perform in the paragraph “Ensemble Combiner” in Section 3.2 and is the reason why $p(q_{(i)})$ should form a valid PMF.
> >
> > Now, for why is it necessary for the coefficients to satisfy $\sum_{i=1}^{d_L}\beta = 1$ in line 261: referring again to the paragraph “Ensemble Combiner” in Section 3.2, the combiner is the average over the discrete model distribution (Equation (3)). In line 261, for the neural network, we start with the combiner and work backwards to identify the subpredictors. However, we are restricted in terms of how we can perform this precisely due to the constraint that we need the weights of the combiner to sum to one (Equation (3)). As we mention around line 261, one way this constraint can be satisfied is if $\sum_{i=1}^{d_L}\beta = 1$.
> >
> > We hope these explanations make the relevance clearer.
> >
> > >Line 276: h_{(1)} depends on the training set, but it is never explicitly mentioned how up to that point. I assume the models are trained, but this should be clearly stated. Are different training sets used since, in the expectations further below (Eq. 10), D is a random variable?
> >
> > **Response (7)**: As stated in Equation (5), $h_{(1)}$ is a hidden node in the first hidden layer: a weighted sum of the inputs followed by an activation function (ReLU in this case). It should, therefore, be implicitly clear that $h_{(1)}$ will depend on the training set through the model weights $w_{(1)}$ in the weighted sum.
> >
> > With respect to different training sets: yes, that is correct, $D$ is the random variable that represents draws of training pairs from the data distribution $P(X,Y)$ to form the sets $\mathcal{D}$ (see Section 3.1). As such, each predictor $q(x) = q(x; \mathcal{D})$ depends on the training set on which it was trained. And the expectation is over all such training sets.

---

> > > ### Author Response · Authors · 2025-11-20
> > >
> > > (continued)
> > >
> > > >What exactly is plotted in Figure 3 (right)? How are all the terms computed?...
> > >
> > > **Response (8)**: In Figure 3 (right), we plot the bias-variance-diversity decompositions for MNIST (top) and CIFAR10 (bottom). That is, we plot both the expected risk as well as the bias, variance, and diversity terms as they are represented in Equation (12).
> > >
> > > To estimate these terms, we need to estimate the expectations over $X, Y$ and $D$ (as in Equations (10) and (12)). With reference to Figure 4 in Appendix B.1: we first perform the usual train-test split of our data. We then use our test set to estimate the expectations over $X$ and $Y$ by taking an average (see Equation (24)).
> > >
> > > We subsample $90\%$ subsets from the training set without replacement to form 50 trial training sets. On each trial set, we train the same neural network (same in the sense that it is initialized with the same seed – see Response (12)). Consequently we have 50 versions of the neural network, each trained on a slightly different training subset. And, in each case, each subpredictor will also depend on the training subset that it was trained on. We then take the average over these 50 training sets (as in Equation (24)).
> > >
> > > We can then proceed to compute (estimate) the classical decomposition of Geman et al. (1992) (which is what is shown in the left hand side of Figure 3) and the new bias-variance-diversity decomposition (the right hand side of Figure 3). So Option 1 is correct: the same training subsets are used in both the left and right hand side of Figure 3.
> > >
> > > # Minor comments
> > >
> > > >Line 256: q^i versus q_{(i)} — why the change from subscript to superscript?
> > >
> > > **Response (9)**: The notation $q_{(i)}$ denotes the $i$-th subpredictor in the implicit ensemble. The notation $q^i$ denotes the vector component of the predictor (note the preceding text that says “at the level of components”).
> > >
> > > >Section 3.1: The indices i are used both for training examples and subpredictors, which causes confusion (e.g., the formula for R_{emp} on line 117 does not make sense). It is also used in d_i, where I assume it refers to input space dimension.
> > >
> > > **Response (10)**: We kindly thank the reviewer for pointing out the typo in the equation for $R_{emp}$, we have fixed this in the updated version.
> > >
> > > With respect to the indices $i$, these indices are simply used as labels and changing the labels does not affect the object (i.e. $q{(i)}$ versus $q_{(j)}$). Thus, on their own, it should be clear what $q_{(i)}$ refers to versus $x^{(i)}$, even if they use the same label $i$. However, when multiple of these appear in the same expression, then it is necessary to change the label (for example, in the case of $R_{emp}$ mentioned).
> > >
> > > >Line 376: "It is known that width is the primary factor for good network performance" — this should be toned down, as for example, large language models often require increased depth rather than just increased width.
> > >
> > > **Response (11)**: This statement was made with respect to the referenced paper by Golubeva et al. (2021). We thank the reviewer for pointing this out as our intention did not come across clearly. We have made the necessary changes to the updated version of the paper.
> > >
> > > >Line 849 (Appendix): "Importantly, we use the same seed to initialize the neural network weights on each trial set." If this is an important point, it should be discussed in the main body: what happens when seeds are not fixed?
> > >
> > > **Response (12)**: Note that we did include it as a footnote (line 431) in the main paper. Like many other papers, we have opted to place most of our experimental details in the appendices. Thus, many important experimental details do not make it to the main paper.
> > >
> > > Per experiment, we fixed the initialization seed for two reasons, to minimize additional sources of randomness (see for example works such as Adlam & Pennington (2020) and D’ Ascoli et al. (2020) which study bias-variance decompositions with randomness coming from weight initialization) and to ensure that sub-predictors can be matched across trials to form the centroid (see for example Appendix C.7). Of course, this does not prevent us from re-running each entire experiment (training 350-450 models) with additional seeds. As the computational cost was excessive, we did not do this.
> > >
> > > We have updated Appendix B.1 to include the above reason.
> > >
> > > >Experiments are a bit toyish. Ideally, it would be interesting to include experiments on more realistic datasets, even just a ResNet on CIFAR-10 rather than fully convolutional neural networks.
> > >
> > > **Response (13)**: We have chosen the CNN architecture due to computational constraints. Recall that, since we train each model width with 50 different random samplings of the training data, 350-450 (9 widths x 50 samplings) models must be trained to produce the graphs in Figure 3. Realistically sized ResNets are therefore prohibitively expensive for such an analysis, and would likely take several weeks or months to train with our resources.

---

> > > > ### Comment · Reviewer_ZfaN · 2025-11-25
> > > >
> > > > Thanks for your response.
> > > >
> > > > After reading the appendix, I now understand what is plotted in Figure 3 from Equations 22 and 24. I think the main document should make it clear that the expectation over X in equation 12 is replaced by an empirical mean, which caused me confusion.
> > > >
> > > > I am not opposed to publication, as I believe the idea is sound and interesting enough to share with the ICLR community. I have raised my score accordingly.
> > > >
> > > > However, I reiterate that the clarity of the paper could be greatly improved. Below are some suggested improvements:
> > > >  - Use notations as in the Deep Learning book (random variables vs. deterministic variables, vectors vs. scalars, etc.).
> > > >  - Use different dummy variable symbols when referring to different objects.
> > > >  - Improve the flow of the development, e.g.,
> > > >     - From Eq. 2 to Eq. 3, p(.) plays a different role.
> > > >     - From Section 3.1 to the last paragraph of Section 3.2, q either outputs an R^d vector or is a density function.
> > > >     - ...

---

> > > > > ### Author Response · Authors · 2025-11-26
> > > > >
> > > > > We kindly thank the reviewer for the change in score.
> > > > >
> > > > > We thank the reviewer for the suggestions to improving the clarity of the paper -- we're busy reviewing the paper with your pointers in mind.

---

### Official Review · Reviewer_XSY4 · 2025-11-01

**Soundness:** 2
**Presentation:** 3
**Contribution:** 2
**Rating:** 4
**Confidence:** 3

**Summary:**

This paper finds that by interpreting single neural networks as implicit ensembles, one can use existing decompositions for the risk of ensembles into a bias, variance and diversity term to explain the expected risk of these single networks. They then argue that the diversity term in this decomposition of the risk of a neural network forms a new source of implicit regularization regulating the variance term. Finally, they empirically investigate this regularization in experiments using CNNs and MLPs.

**Strengths:**

The paper makes use of the elegant idea to interpret single neural networks as implicit ensembles, thereby being able to make use of already developed theory for ensembles to analyze single networks. Using this approach, they can in particular identify the diversity of the subpredictors of the model as a new source of implicit regularization, which is a relevant finding.
More generally, the story of the paper was relatively clear and the paper was well-organized. Using the examples of the square loss and the KL divergence throughout the paper helped a lot in being able to understand the introduced concepts better.

**Weaknesses:**

One concern I have is that the paper does not provide substantially new insights beyond applying the existing theory from Wood et al. (2023) to subpredictors of a single neural network. I would have been interested in seeing slightly more discussion on where they provide new insights beyond the results from Wood et al. (2023) throughout the paper.
Furthermore, most of the discussion in Section 4.4. seemed relatively speculative (e.g., multiple 'we hypothesize') and I would have been interested in seeing these hypotheses being empirically tested more explicitly.

**Questions:**

1. I assume that the Bias-variance-diversity decomposition (Theorem 1) does not need independence between the subpredictors (since this would not be fulfilled for the subpredictors in your model). Could you expand on why independence is not necessary here and how non-independence affects the different subterms?
2. How are the Bias and Variance terms in your decomposition in Equation (10) and (12) related to a more classical Bias-Variance Decomposition?
3. Could you clarify in more detail which points mentioned in the discussion in Section 4.4. you think have been validated by your experiments and how you would test any other hypotheses that remain?
4. Do you think the following is generally true (even for subpredictors in large networks):
> On their own, these subpredictors are relatively simple models (a hidden node multiplied by a weight) and each subpredictor will likely have a high bias error.
5. Could you explain what you mean by this sentence in more detail/why you made this choice:
> To minimize some of the implicit regularization effects due to mini-batch SGD (Smith et al., 2021), we make use of full batch gradient descent as an optimizer.
6. What is your explanation for why the diversity term is tracking the variance term so closely (e.g., in Figure 1)?

---

> ### Author Response · Authors · 2025-11-20
>
> We thank the reviewer for the comments and the questions. We appreciate that the reviewer found the identification of the diversity of the subpredictors as a new source of implicit regularization to be a relevant finding.
>
> Below we address the concerns raised, as well as the questions:
>
> # Weaknesses
>
> >One concern I have is that the paper does not provide substantially new insights beyond applying the existing theory from Wood et al. (2023) to subpredictors of a single neural network.
>
> **Response (1)**: We would like to emphasize the new insights that we have gained beyond Wood et al. (2023):
> * The discovery of a new source of implicit regularization in a single neural network.
> * A novel formulation of a neural network as an implicit ensemble.
> * An interesting connection between neural network parameterizations and our implicit ensemble formulation.
>
> We have added a ‘Future work’ paragraph (line 492 in the latest version) which provides further possible insights. See also our Response (1) for Reviewer qzCe.
>
> >Furthermore, most of the discussion in Section 4.4. seemed relatively speculative (e.g., multiple 'we hypothesize') and I would have been interested in seeing these hypotheses being empirically tested more explicitly.
>
> **Response (2)**: For Section 4.4, we have added additional sentences to clarify which hypotheses were validated and where. We have also added an additional paragraph in Section 5 which links back to the hypotheses made in Section 4.4 (line 468 in the latest version). Finally, we have also added a ‘Future work’ paragraph (line 492 in the latest version) which discusses ideas that we did not investigate further in our paper.
>
> See also our Response (5) for more detail.
>
> # Questions
>
> >Could you expand on why independence is not necessary here and how non-independence affects the different subterms?
>
> **Response (3)**: Mathematically, as far as the proof of Theorem 1 is concerned, no assumption on independence of the subpredictors in the ensemble is necessary (see Appendix A.2.2). Furthermore, thinking of an explicit ensemble, there exists statistical dependencies between the outputs of ensemble members (for example, correlation between the outputs of ensemble members is sometimes used as a proxy for diversity). In fact, the diversity term in the decomposition accounts for statistical dependencies between ensemble members (Wood et al, 2023).
>
> See also Response (8).
>
> >How are the Bias and Variance terms in your decomposition in Equation (10) and (12) related to a more classical Bias-Variance Decomposition?
>
> **Response (4)**: In the classical case, the bias measures the distance of a single estimator from the variable to be estimated. The same is true in Equations (10) and (12) – only there is a sum of these bias terms over the subpredictors.
>
> The same holds for the variance term.
>
> >Could you clarify in more detail which points mentioned in the discussion in Section 4.4. you think have been validated by your experiments and how you would test any other hypotheses that remain?
>
> **Response (5)**:
> * “Overfitting” hypothesis: For the main paper, we showed that this holds for classifications tasks on MNIST and CIFAR10 with two architectures (MLP and CNN) – these results are summarized in Figures 1 and 3. In the appendices, we also empirically validate the hypothesis on additional experiments.
>
> * “Capacity” claim: For shallow (or not very deep) networks (see Response (6) below), we empirically see high bias error in the same experiments mentioned above. See for example Figure 3.
>
> * “Width” hypothesis: This is the only claim that we have not empirically validated. It can potentially be tested by following the methods of Golubeva et al. (2021) and, for networks with the same number of parameters but different widths, compute the contribution from the diversity term.
>
> >Do you think the following is generally true (even for subpredictors in large networks):
> On their own, these subpredictors are relatively simple models (a hidden node multiplied by a weight) and each subpredictor will likely have a high bias error.
>
> **Response (6)**: No, this is generally not true – we originally made this statement with respect to Figure 2 for the case of a shallow network. We have amended the sentence in the updated version to reflect this.

---

> > ### Author Response · Authors · 2025-11-20
> >
> > (continued)...
> >
> > >Could you explain what you mean by this sentence in more detail/why you made this choice:
> > To minimize some of the implicit regularization effects due to mini-batch SGD (Smith et al., 2021), we make use of full batch gradient descent as an optimizer.
> >
> > **Response (7)**: We were interested in this choice for two reasons:
> >
> > * There are other sources of implicit regularization that have been studied apart from diversity. One such example is due to mini-batch SGD – notably, when considering the loss function for the infinitesimal version of gradient descent, the discretizing and random batching of SGD modifies the loss function with two penalty terms (in other words, implicit regularization) (Smith et al., 2021). Motivated to study the strength of the diversity regularizer in our paper, we wanted to (at least partly) reduce the strength of the implicit regularization due to SGD. Consequently, we were motivated to perform this experiment using full batch training (which removes the regularizing effect of the second penalty term).
> > * We were also interested in the possible effect that batching (or lack there of) may have on the diversity of the subpredictors, as measured by the diversity term.
> >
> > For further details, we refer to Appendix C.1.
> >
> > >What is your explanation for why the diversity term is tracking the variance term so closely (e.g., in Figure 1)?
> >
> > **Response (8)**: This is an excellent question. Why this occurs is an open question and one that we are currently investigating. However, we can provide some intuition.
> >
> > For simplicity, consider the square loss and the decomposition given in Equation (10). Consider the following extreme case: suppose that all $q_{(i)}(x) = \bar{q}(x)$ (in other words, all subpredictors are identical) and that for all $q_{(i)}(x; \mathcal{D}) = q_{(i)}(x)$ (in other words, each data set $\mathcal{D}$ gives the same subpredictor). Then, in this case, both the diversity term and the variance term vanishes.
> >
> > Now suppose that, for some $i$ and $j$, we perturb the two subpredictors as follows: $q_{(i)}(x) = \bar{q}(x) \rightarrow q_{(i)}(x) = \bar{q}(x) + \delta(\mathcal{D})$ and $q_{(j)}(x) = \bar{q}(x) \rightarrow q_{(j)}(x) = \bar{q}(x) - \delta(\mathcal{D})$. Then, the combiner $\bar{q}$ is still independent of $\mathcal{D}$ (since, for square loss the combiner is a simple average, the delta’s cancel). This perturbation turns on the variance term (most of its terms vanish):
> >
> > \begin{equation}
> > Variance = \frac{2}{d_L}\mathbb{E}_{D}[(\mathbb{E}_D[\delta(\mathcal{D})] - \delta(\mathcal{D}))^2].
> > \end{equation}
> >
> > Similarly,
> >
> > \begin{equation}
> > Diversity = \frac{2}{d_L}\mathbb{E}_{D}[\delta^2(\mathcal{D})].
> > \end{equation}
> >
> > Hence, as we perturb some of the subpredictors, not only does the variance error turn on, but so does the diversity, with both terms dependent on the perturbation. We can imagine perturbing more subpredictors and generating more such terms within a neural network.
> >
> > Furthermore, for the example above, if we now add more or remove $q_{(i)}(x) = \bar{q}(x)$ to the ensemble, then we change $d_L$ (the width). Both of the above terms will change proportionally through their dependence on $d_L$ and, as capacity (width) varies, appear to track each other.
> >
> > Although the above case is rather unrealistic and extreme, it does provide some intuition as to why the diversity tracks the variance.

---

> > ### Comment · Reviewer_XSY4 · 2025-11-28
> >
> > I thank the authors for answering my questions, addressing my concerns, and making updates to the paper.
> >
> > My remaining concerns are:
> > (1) I still believe that the primary contribution of the paper is the application of the results from Wood et al. (2023) to subpredictors of single neural networks. The other two contributions the authors listed seem to follow naturally from this main contribution
> > (2) In Section 4.4, the paragraph about 'Capacity' still does not make clear enough that these statements only hold for shallow networks. Furthermore, the hypotheses formulated in the paragraph about 'Width' seem important to verify rather than merely hypothesize about them.
> >
> > I will maintain my score.

---

> > > ### Author Response · Authors · 2025-11-28
> > >
> > > We thank the reviewer for the feedback. While we do acknowledge that the reviewer wishes to maintain their score, we would like to provide additional comments addressing their remaining concerns.
> > >
> > > >I still believe that the primary contribution of the paper is the application of the results from Wood et al. (2023) to subpredictors of single neural networks. The other two contributions the authors listed seem to follow naturally from this main contribution
> > >
> > > **Response (9)**: To state things from a different perspective: while Wood et al’s (2023) work is highly important to our results, our work provides a concrete connection between two seemingly disparate fields: the field of ensemble theory and the field of implicit regularization in neural networks. Apart perhaps from Olson et al (2018), we believe that this connection has been highly unexplored and that our work makes a nontrivial contribution towards this. This connection is our primary contribution – a previously unseen implicit regularizer in a neural network that is exposed thanks to recent new results in the seemingly unrelated field of the theory of ensembles. Thus, while we acknowledge and value the reviewer’s criticism, we do believe that it under emphasizes the value of this new connection.
> > >
> > > >In Section 4.4, the paragraph about 'Capacity' still does not make clear enough that these statements only hold for shallow networks. Furthermore, the hypotheses formulated in the paragraph about 'Width' seem important to verify rather than merely hypothesize about them.
> > >
> > > **Response (10)**: To further address this, we have made the following amendments to the paper:
> > > * We have changed ‘Capacity’ to 'Shallow Network Capacity' for further emphasis.
> > > * Since it is somewhat tangential to the goals of the paper, we have removed the paragraph on Width. We therefore only mention this aspect in the Future Work section.

---

### Official Review · Reviewer_iryq · 2025-11-01

**Soundness:** 3
**Presentation:** 3
**Contribution:** 3
**Rating:** 6
**Confidence:** 2

**Summary:**

This work reinterprets a single neural network as an implicit ensemble, drawing upon existing literature on parameterizations that incorporate output (or logit) scaling factors, as well as on diversity theory.

**Strengths:**

- Section 3 provides a well-organized summary of the essential components from Wood et al. (2023), which helps readers grasp the necessary background.
- It is quite interesting to incorporate the output (or logit) scaling factors from MFP, SP, or muP into the ensemble combiner. This reminds me of Kirsch et al. (2025), where a slight connection between implicit ensembles and the NTK was discussed.
- I enjoyed the “ensemble” decomposition of a “single” model presented in Section 4.3. In the past, there have been discussions in the context of CNNs suggesting that the final average pooling layer could be interpreted as a form of implicit ensemble; the formulation here feels much more direct and follows the well-defined diversity decomposition of Wood et al. (2023).

---
- Kirsch et al. (2025), (Implicit) Ensembles of Ensembles: Epistemic Uncertainty Collapse in Large Models.

**Weaknesses:**

- The limitation, as also acknowledged by the authors in Appendix A.1, is that the analysis is restricted to networks whose final component is an MLP with ReLU activations. In practice, modern neural network architectures (yes, there’s really only one nowadays, transformers) do not typically fit this assumption, which makes this a clear weakness of the work. That said, it still offers a valuable perspective, so I wouldn’t consider this a major flaw.

**Questions:**

- wrong left quotation mark in line 89.

- One notable point in Wood et al. (2023) is that they consider (pre-softmax) logit ensembling for classification models. I’ve often felt that this differs somewhat from the common practice of performing (post-softmax) probability ensembling. While logit ensembling is sometimes used, from a Bayesian perspective, if we think of modeling the categorical predictive distribution, (post-softmax) probability ensembling would be the more appropriate formulation in the context of Bayesian model averaging. I wonder, though, whether a similar line of reasoning could be extended to justify post-softmax ensembling as well; what are your thoughts on that?

- One interesting aspect of neural network ensembles is that training individual members separately and then combining them often leads to different outcomes compared to training them jointly in an ensemble form from the start (Allen-Zhu and Li, 2023). The ensemble considered in this work corresponds to the latter case. I wonder whether similar results would still hold if the former approach were taken instead.

---
- Allen-Zhu and Li (2025), Towards Understanding Ensemble, Knowledge Distillation and Self-Distillation in Deep Learning.

---

> ### Author Response · Authors · 2025-11-20
>
> We thank the reviewer for the comments and the questions. We appreciate that the reviewer enjoyed our method of decomposing a neural network into an ensemble of members, as well as finding the inclusion of the parameterization scaling factor into the ensemble combiner interesting. We also appreciate that the reviewer found Section 3 to be a well-organized summary of the key results of Wood et al. (2023).
>
> Below we address the concerns raised, as well as the questions:
>
> # Weaknesses
>
> >The limitation, as also acknowledged by the authors in Appendix A.1, is that the analysis is restricted to networks whose final component is an MLP with ReLU activations. In practice, modern neural network architectures (yes, there’s really only one nowadays, transformers) do not typically fit this assumption, which makes this a clear weakness of the work. That said, it still offers a valuable perspective, so I wouldn’t consider this a major flaw.
>
> **Response (1)**: This is a valid point, we hope to extend the results to transformers in the future (we have added a ‘Future work’ paragraph mentioning this (line 492) in the latest version). We do appreciate that the reviewer finds this to not be a major flaw.
>
> # Questions
>
> >wrong left quotation mark in line 89.
>
> **Response (2)**: Thank you – we have fixed it in the current version.
>
> >One notable point in Wood et al. (2023) is that they consider (pre-softmax) logit ensembling for classification models. I’ve often felt that this differs somewhat from the common practice of performing (post-softmax) probability ensembling. While logit ensembling is sometimes used, from a Bayesian perspective, if we think of modeling the categorical predictive distribution, (post-softmax) probability ensembling would be the more appropriate formulation in the context of Bayesian model averaging. I wonder, though, whether a similar line of reasoning could be extended to justify post-softmax ensembling as well; what are your thoughts on that?
>
> **Response (3)**: We thank the reviewer for the very interesting comments and question. To be sure, by “whether a similar line of reasoning could be extended to justify post-softmax ensembling as well”, did you perhaps mean pre-softmax?
>
> >One interesting aspect of neural network ensembles is that training individual members separately and then combining them often leads to different outcomes compared to training them jointly in an ensemble form from the start (Allen-Zhu and Li, 2023). The ensemble considered in this work corresponds to the latter case. I wonder whether similar results would still hold if the former approach were taken instead.
>
> **Response (4)**: This is a highly interesting line of thought. If one considers a deep MLP, for example, then in our formulation of the implicit ensemble, the subpredictors share weights among themselves. In this case, it therefore does not seem possible to train the individual members (subpredictors) independently from one another.
>
> However, consider a single hidden layer MLP as in Figure 2. Conceptually, this is perhaps the closest case of a single neural network that resembles an explicit ensemble: for the least-squares loss, the members of the ensemble are simple hidden nodes multiplied by an outgoing weight and, moreover, there is no weight sharing amongst the members. The combiner, for least squares loss, is then an average. Consequently, one could attempt to train them independently from one another and then combine them at the end using an average.
>
> In Allen-Zhu and Li (2023) and for the case of an ensemble of neural networks, the members of the ensemble are trained independently on the same data set using the same training algorithm, the difference being weight initialization and possible randomness introduced by the training algorithm. Interestingly, the implicit ensemble view of the single hidden layer MLP discussed above shares this setting except for the case of the data set being the same; most notably, due to the ReLU activation, nodes exhibit activation patterns and these activation patterns divide the input space into activation regions. Consequently, a given member of the implicit ensemble sees a ‘different’ data set when compared to the other ensemble members (the same holds true when training them using the latter approach in the reviewer’s comment).
>
> Would similar results still hold if the former approach were taken: for the single hidden layer MLP, we strongly suspect that training the subpredictors of the MLP using the former method versus the latter method would lead to different outcomes. However, due to the activation regions mentioned above, the exact nature of the difference in the outcomes may deviate from those investigated in Allen-Zhu and Li (2023). We believe such an investigation to be a fascinating avenue of research. Furthermore, while the above holds for ReLU activations, we do wonder what the consequences would be for other activations (such as sigmoid).

---

> > ### Comment · Reviewer_iryq · 2025-11-21
> >
> > >> One notable point in Wood et al. (2023) is that they consider (pre-softmax) logit ensembling for classification models. ...
> > >
> > > Response (3): We thank the reviewer for the very interesting comments and question. To be sure, by “whether a similar line of reasoning could be extended to justify post-softmax ensembling as well”, did you perhaps mean pre-softmax?
> >
> > Upon reflection, I realize this work interprets the single model as an implicit ensemble; it fundamentally precludes the need to consider post-softmax ensembling, as the implicit ensemble is intrinsically realized through logit ensembling. My apologies for the confusion.

---

> > > ### Author Response · Authors · 2025-11-26
> > >
> > > No problem at all -- thanks for letting us know.

---

### Official Review · Reviewer_qzCe · 2025-11-03

**Soundness:** 2
**Presentation:** 3
**Contribution:** 2
**Rating:** 4
**Confidence:** 3

**Summary:**

- This work combines two lines of work on understanding implicit regularization in deep neural networks and studying ensembles from the perspective of bias, variance and diversity terms. This work considers a single neural network as an ensemble of multiple neural networks and hence, breaks down the loss like ensembles, They connect the diversity term to an additional form of implicit regularization in neural networks. They also show that this diversity term is large for overparmeterized neural networks and can explain the double decent framework.

**Strengths:**

- The paper overall is well written.
- The paper combines two different lines of work on ensembles and implicit regularization in NNs. I find the study of implicit regularization through the ensemble loss decomposition interesting and novel.

**Weaknesses:**

- Although the work provides interesting connections, the main theorem 1 is taken from previous work and does not provide new theoretical contribution.
- In this work, the authors have considered one way of decomposing the model into multiple subnetworks along the last layer. But, would the results change if we use a different decomposition?

**Questions:**

- In this work, the authors have considered one way of decomposing the model into multiple subnetworks along the last layer. But, would the results change if we use a different decomposition?
- It seems like in all the experiments, diversity has the same behavior as variance. Can the authors suggest a case where this is not true?
- Does this diversity has any connections to any other form of implicit regularization that is classically studied in this literature?

---

> ### Author Response · Authors · 2025-11-20
>
> We thank the reviewer for the comments and the questions. We appreciate that the reviewer finds the paper overall well written and the recognition of the novelty concerning implicit regularization through loss decomposition.
>
> Below we address the concerns raised, as well as the questions:
>
> # Weaknesses
>
> >Although the work provides interesting connections, the main theorem 1 is taken from previous work and does not provide new theoretical contribution.
>
> **Response (1)**: Our main theoretical contribution was to demonstrate a new source of implicit regularization in neural networks, which we do in two steps: (1) We first derive a precise decomposition of a neural network into an ensemble of subpredictors in such a way that it is possible to apply Theorem 1 from Wood et al (2023).  (Without this precise decomposition, the application of Theorem 1 to a single neural network would not be possible.). This derivation is not trivial in the general case where different network parameterizations are possible, and either regression or classification are considered. As far as we are aware, this structure within the network (Tables 1 and 2) is unique to our derivation. (2) We then apply Wood et al. (2023) in order to explain the regularization effect of subpredictor diversity - a new theoretical results that we also explore empirically.
>
> Under our approach, we also demonstrate a new theoretical connection between certain parameterizations of the neural network and the implicit ensemble framework. For example, MFP emerges as the more natural form for the implicit ensemble. (See the discussion around Equation (7) and the Discussion paragraph after Table 2).
>
> >In this work, the authors have considered one way of decomposing the model into multiple subnetworks along the last layer. But, would the results change if we use a different decomposition?
>
> **Response (2)**: Yes, the results would change in the following sense: we would likely not be able to apply the theoretical framework of Wood et al. (2023) to the neural network and the resulting decomposition of the risk (Equations 10 and 12) would not follow (and, hence, the diversity term would remain hidden in the network’s risk decomposition). We draw emphasis to the fact that, if we are to apply the framework of Wood et al. (2023) to a single neural network, we are constrained in terms of how we construct the implicit ensemble of the neural network. Briefly (further details may be found in Section 4.2), the neural network output fixes the combiner. The combiner is a weighted sum of subpredictors, where the weights are nonnegative and need to sum to unity (See Section 3.2). Hence, we are restricted in terms of how we decompose the network into subnetworks (subpredictors). A more arbitrary decomposition of the network into subnetworks will likely violate these constraints.
>
> # Questions
>
> >In this work, the authors have considered one way of decomposing the model into multiple subnetworks along the last layer. But, would the results change if we use a different decomposition?
>
> **Response (3)**: We kindly refer the reviewer to Response (2) above.
>
> >It seems like in all the experiments, diversity has the same behavior as variance. Can the authors suggest a case where this is not true?
>
> **Response (4)**: This is an interesting question – currently, we have not observed this in our experiments. However, it’s possible one may see such behavior when extending the framework to other networks than what we considered in the paper (networks with a FC layer at the end).
> Since it’s related, see also Response (8) for Reviewer XSY4.
>
> >Does this diversity has any connections to any other form of implicit regularization that is classically studied in this literature?
>
> **Response (5)**: As far as we are aware, no, the diversity term as an implicit regularizer does not have any direct connection to other forms of implicit regularization in the literature. The closest work we could identify was Olson et al. (2018) – see our discussion in the Related Work section (Section 2) – that argue that there may be some form of implicit regularization due to ensemble-like effects in a neural network.

---

> > ### Author Response · Authors · 2025-11-28
> >
> > We would like to kindly remind the reviewer of our rebuttal above.
> >
> > In case our rebuttal has addressed all your concerns, we would appreciate an increase in score.

---

### Author Response · Authors · 2025-12-03

# Post-Score Reversion Message to the AC
We first wish to thank the reviewers for their questions and criticisms which, as with most rebuttal periods, we believe have improved the paper.

**Our Result:** The principal idea of our paper connects two seemingly disparate fields: the field of *ensemble theory* and the field of *implicit regularization in neural networks*. This connection is highly under explored in the deep learning literature. The primary result in our work contributes to this connection in the following way: using recent results from Wood et al. (2023) on ensemble theory, we demonstrate that a single neural network’s expected risk decomposition contains a hidden term called *diversity*. Moreover, this diversity term acts as an implicit regularizer. We demonstrate the regularization effect of the diversity term on both MLPs and CNNs and over many different training configurations. Thus, we provide a concrete connection between the two mentioned fields.

To achieve this required the application of Wood et al.’s (2023) results for ensembles to a single neural network. This required us to introduce a novel decomposition of the neural network as an implicit ensemble of subpredictors under constraints imposed by Wood et al’s (2023) framework. In doing so, we also make an interesting connection to different parameterizations of a neural network, with some parameterizations already being in a ‘natural’ form of an implicit ensemble without the need for factorization of the network. Our work opens many new avenues of research, some of which we detail in our Future Work section in the most recent versions of the paper.

**Rebuttal:** During the rebuttal phase, we were able to answer all technical questions, and update the paper as requested by the reviewers. Reviewers were positive about the soundness and novelty of the paper. Our initial score for the paper started at 6 4 4 2. After some discussion, we were able to increase our score to 6 4 4 6, with discussions with other reviewers still ongoing. Although the recent information leak has invalidated this increase in score, we still find it pertinent to point out as the initial scores reflect misunderstandings that are corrected during the rebuttal period. Overall, two reviewers motivated for acceptance and no reviewers seemed strongly opposed to publication.

**Summary:** In summary, we believe that our work provides novel and interesting results to ICLR 2026 and to the deep learning community in general.

---

### Meta-Review · Area_Chair_7qhT · 2026-01-07

**Summary:**

The reviewers agree that the paper offers an interesting and technically sound connection between ensemble theory and implicit regularization in single neural networks, exposing a diversity term that behaves like an implicit regularizer.

all reviewers acknowledge that the idea is relevant while they mainly concerned about:

- Novelty compared with Wood et al. (2023),

- Limited experimental scope (MLPs and small CNNs)

- Some clarity and notation problems

The scores are quite mixed (6 4 4 6 after the rebuttal according to the summary of the authors), suggesting this is a borderline paper.

**Reviewer Concerns:**

**Addressed in the rebuttal**

- The authors clarified how their contribution goes beyond Wood et al. (2023).

- Multiple writing and technical questions were resolved, e.g., how diversity and variance relate and the role of centroids/combiner

**Still outstanding**

- Concerns on novelty relative to Wood et al. (2023) remain

- Experiments remain relatively small scale and are limited to networks with ReLU MLP

- Some clarify issues remain

**Reviewer Scores:**

Reviewer qzCe: remains at 4 and didn't reply to the rebuttal. I feel that they might have not significantly improved the score as the main concern is on the novelty.

Reviewer iryq: originally gave 6 but didn't reply to the rebuttal.

Reviewer XSY4: explicitly states they will remain at 4.

Reviewer ZfaN: raised from 2 to 6 according to the authors.

---

### Decision · Program_Chairs · 2026-01-26

Reject